# The Slingshot phosphatase 2 is required for acrosome biogenesis during spermatogenesis in mice

Ke Xu[1,2,3,4†], Xianwei Su[5†], Kailun Fang[6†], Yue Lv[5,7], Tao Huang[1,2,3,4], Mengjing Li[1,2,3,4], Ziqi Wang[1,2,3,4], Yingying Yin[1,2,3,4], Tahir Muhammad[1,2,3,4‡], Shangming Liu[8], Xiangfeng Chen[9], Jing Jiang[10], Jinsong Li[10], Wai-Yee Chan[1,5], Jinlong Ma[1,2,3,4], Gang Lu[1,5*], Zi-Jiang Chen[1,2,3,4,5,7,9,11*], Hongbin Liu[1,2,3,4,5,11*]

[1]Center for Reproductive Medicine, Shandong University, Jinan, China; [2]Key laboratory of Reproductive Endocrinology of Ministry of Education, Shandong University, Jinan, China; [3]Shandong Provincial Clinical Medicine Research Center for Reproductive Health, Shandong University, Jinan, China; [4]Shandong Technology Innovation Center for Reproductive Health, Shanghai, China; [5]CUHK-SDU Joint Laboratory on Reproductive Genetics, School of Biomedical Sciences, Chinese University of Hong Kong, Hong Kong, China; [6]Institute of Neuroscience, State Key Laboratory of Neuroscience, CAS Center for Excellence in Brain Science and Intelligence Technology, Chinese Academy of Sciences, Shanghai, China; [7]Shandong Key Laboratory of Reproductive Medicine, Shandong Provincial Hospital, Affiliated to Shandong First Medical University, Jinan, China; [8]School of Basic Medical Sciences, Shandong University, Jinan, China; [9]Shanghai Key Laboratory for Assisted Reproduction and Reproductive Genetics, Shanghai, China; [10]Genome Tagging Project (GTP) Center, Shanghai Institute of Biochemistry and Cell Biology, Center for Excellence in Molecular Cell Science, Chinese Academy of Sciences, Shanghai, China; [11]Research Unit of Gametogenesis and Health of ART-Offspring, Chinese Academy of Medical Sciences, Shanghai, China

*For correspondence:
lugang@cuhk.edu.hk (GL);
chenzijiang@hotmail.com (Z-JiangC);
hongbin_sduivf@aliyun.com (HL)

†These authors contributed equally to this work

Present address: ‡Department of Cell Biology and Anatomy, New York Medical College, Valhalla, United States

**Abstract** The acrosome is a membranous organelle positioned in the anterior portion of the sperm head and is essential for male fertility. Acrosome biogenesis requires the dynamic cytoskeletal shuttling of vesicles toward nascent acrosome which is regulated by a series of accessory proteins. However, much remains unknown about the molecular basis underlying this process. Here, we generated *Ssh2* knockout (KO) mice and HA-tagged *Ssh2* knock-in (KI) mice to define the functions of Slingshot phosphatase 2 (SSH2) in spermatogenesis and demonstrated that as a regulator of actin remodeling, SSH2 is essential for acrosome biogenesis and male fertility. In *Ssh2* KO males, spermatogenesis was arrested at the early spermatid stage with increased apoptotic index and the impaired acrosome biogenesis was characterized by defective transport/fusion of proacrosomal vesicles. Moreover, disorganized F-actin structures accompanied by excessive phosphorylation of COFILIN were observed in the testes of *Ssh2* KO mice. Collectively, our data reveal a modulatory role for SSH2 in acrosome biogenesis through COFILIN-mediated actin remodeling and the indispensability of this phosphatase in male fertility in mice.

## Editor's evaluation

This important study reports convincing data supporting the essential role of SSH2 in the regulation of acrosome biogenesis during spermiogenesis. The conclusion is well supported by the

experiments performed both in vivo and in vitro. This work will help understand the molecular process of sperm assembly and also identify potential genetic mutations responsible for acrosomal defects in male infertility patients.

## Introduction

The most common cause of non-obstructive azoospermia is impaired spermatogenesis (i.e., the production of mature spermatozoa), which is behind about 15% of infertility cases in men (*Agarwal et al., 2021*). In basic studies of mice, spermatogenesis—a continuous course to generate spermatozoa within the seminiferous epithelia incorporating the mitosis of undifferentiated spermatogonia, meiosis of spermatocytes, and eventually spermatid cytodifferentiation—is routinely subdivided into 12 stages of developmental progression (*Meistrich and Hess, 2013*; *Nakata et al., 2015*; *Oakberg, 1956*). Note that the sub-process through which post-meiotic spermatids develop into mature spermatozoa is termed 'spermiogenesis' (*Bao and Bedford, 2016*), which is marked by a series of cellular remodeling events that include chromatin condensation, flagellum formation, elimination of redundant cytoplasm, and biogenesis of the acrosome—an acidic, membranous organelle positioned over the anterior part of the sperm nucleus that functions in fertilization of the egg (*Moreno and Alvarado, 2006*). The transformation of round spermatids involves the cytoskeletal system, especially microfilaments (*Soda et al., 2020*), which shuttles protein and vesicles to the nascent acrosome via dynamic cytoskeletal remodeling orchestrated by various accessory proteins such as the actin-binding protein PROFILIN-3 (*Umer et al., 2021*). Notwithstanding some advancements in acrosome biogenesis have been achieved, much of the molecular basis underlying the engagement of actin organization orchestration in this process still need to be investigated.

Typically, the formation of the acrosome, which is characterized by the generation, trafficking, and fusion of vesicles that harbor the essential acrosomal components such as hydrolytic enzymes and proteases involved in the acrosome reaction, occurs simultaneously with the process of 16-step spermatid differentiation (*Teves et al., 2020*) that has been well documented morphologically. According to the classical descriptions of acrosome biogenesis that follows four sequential phases (the Golgi phase, cap phase, acrosome phase, and maturation phase) (*Clermont and Leblond, 1955*), the nascent acrosome is sequentially assembled from the so-called proacrosomal vesicles that originate from the Golgi apparatus through the biosynthetic pathway (*Khawar et al., 2019*). However, recent experimental findings have implicated some other cellular components (e.g., the endocytic machinery) in vesicular trafficking toward the growing acrosome (*Berruti and Paiardi, 2015*; *Li et al., 2006*), supporting the existence of an extra-Golgi supply of proacrosomal vesicles. Notably, the role of filamentous actin (F-actin) as a cytoskeletal transport platform for proacrosomal vesicles has been documented (*Kierszenbaum et al., 2003b*). In addition, several Golgi-associated proteins are known to function in the transport/fusion of proacrosomal vesicles, including Golgi-associated PDZ and coiled-coil motif-containing protein (GOPC) (*Yao et al., 2002*), protein interacting with C kinase 1 (PICK1) (*Xiao et al., 2009*), and Golgi matrix protein 130 (GM130) (*Han et al., 2017*). More recently, the contribution of the autophagic machinery in Golgi-derived proacrosomal vesicle trafficking has been identified by germ cell-specific *Atg7*-KO (*Wang et al., 2014b*) and *Sirt1*-KO (*Liu et al., 2017*) mouse models, offering new insights into the molecular mechanisms through which the acrosome develops.

As an actin-binding protein, COFILIN is widely known for its cutting and depolymerizing functions that promote the subsequent remodeling of actin filaments (*Wioland et al., 2017*), which is required for many actin-driven events such as mitosis (*Amano et al., 2002*) and organelle trafficking (*Cichon et al., 2012*). Intriguingly, COFILIN was also found to stimulate actin nucleation at a high COFILIN/actin concentration ratio, which in turn favors actin filament assembly (*Andrianantoandro and Pollard, 2006*). It follows that the complex modulation of COFILIN activity, consisting of post-translational modifications, protein binding, and redox reactions (*Namme et al., 2021*), enables the proper rearrangement of F-actin organization in response to various extracellular stimuli. Among these regulatory mechanisms, the phosphorylation/dephosphorylation at the Ser-3 residue stands out as the most crucial. Phosphorylation-mediated inactivation by LIM domain kinases (LIMKs) prevents COFILIN's actin remodeling functions (*Arber et al., 1998*), whereas activating dephosphorylation of COFILIN is catalyzed by phosphatases such as chronophin (*Gohla et al., 2005*) and the Slingshot (SSH)

family of proteins (*Niwa et al., 2002*). Several studies have reported the cofilin phospho-regulation of diverse cellular processes including cell movement (*Nishita et al., 2005*), cytokinesis (*Kaji et al., 2003*), and germ cell development (*Takahashi et al., 2002*).

The ssh gene was originally identified in *Drosophila* mutants with the bifurcation phenotype of bristles and hairs (*Niwa et al., 2002*), and the mammalian homologs encode a set of COFILIN phosphatases, namely the SSH phosphatases. Some biological relevance of SSHs has been revealed, especially the most well-characterized isoform SSH1 (*Bielig et al., 2014*), while much less is known about SSH2 other than its role as a regulator of actin remodeling during neutrophil chemotaxis (*Xu et al., 2015*). According to a gene-expression study focusing on the SSHs, relatively higher expression of SSH2 was observed in testes compared to other organs in mice (*Ohta et al., 2003*). Moreover, in the light of a recent proteomic study which is aiming at the prediction of meiosis-essential genes, SSH2 exhibited increasing protein abundance changes with the proceeding of murine spermatogenesis (*Fang et al., 2021*), illustrating its potential role in male germ cell development. By generating mice carrying targeted knockout (KO) of *Ssh2* and knock-in (KI) mice expressing hemagglutinin (HA)-tagged SSH2, we found that SSH2 is an activator of COFILIN-mediated actin cytoskeleton remodeling and is essential for proper acrosome biogenesis during spermatogenesis and thus required for male fertility. The impaired acrosome biogenesis in *Ssh2* KO mice is likely a consequence of abnormal vesicular trafficking/fusion that can be attributed to disrupted F-actin remodeling.

## Results

### *Ssh2* is essential for male fertility

To explore the role of SSH2 in male fertility, we generated mice with *Ssh2* KO lacking exon 8 using CRISPR/Cas9 genome editing (*Figure 1A*). We confirmed the *Ssh2* KO status in mice using PCR of tail-derived genomic DNA (*Figure 1—figure supplement 1*) and western blotting (WB) of whole testicular lysates. SSH2 was immunodetected in the testes of wild-type (WT) mice but was absent in the *Ssh2* KO samples (*Figure 1B*). Regarding reproductive functional activity, *Ssh2* KO male mice were found to be completely infertile (*Figure 1C*). Six mice of each genotype were examined for fertility test for 2 months. The data indicated infertile status as no pups were obtained when adult *Ssh2* KO males were mated with WT fertile females (8 weeks of age). We examined no differences in the size of the testis, body weight, testis weight, or testis-to-body weight ratio in *Ssh2* KO male mice compared to their WT littermates (*Figure 1D–G*).

Histological examination of WT and *Ssh2* KO testes and epididymides by hematoxylin staining revealed impaired spermatogenesis in *Ssh2* KO males, with no mature spermatozoa observed in the epididymal lumen of *Ssh2* KO mice (*Figure 1H*), that was also confirmed by the sperm count in the cauda of the epididymis (*Figure 1I*). The seminiferous tubules of WT testes were full of spermatogenic cells and contained a basal population of spermatogonia, spermatocytes, and spermatids. In contrast, almost no elongating or elongated spermatids were observed in the testes of *Ssh2* KO males; instead, round spermatid aggregates were found to form a giant multinucleated cell that were further verified to be spermatid clusters by fluorescence-activated cell sorting (FACS) assessment (*Figure 1H*, *Figure 1—figure supplement 2*). Thus, our data suggest that *Ssh2* is essential for male fertility.

### *Ssh2* KO mice exhibit spermatogenic arrest at the early spermatid stage and display aberrantly high germ cell apoptosis

To determine the time point of spermatogenic arrest in *Ssh2* KO males, we first analyzed spermatocyte development by examining chromosomal synapsis during meiotic progression in chromosome spreads from 3-week-old testes of WT and *Ssh2* KO mice. By immunofluorescence co-staining of synaptonemal complex protein 1 (SYCP1) and SYCP3 in the nuclei of spermatocyte spreads—which comprise the central and lateral elements of the synaptonemal complex, respectively (*de Vries et al., 2005*; *Yuan et al., 2000*)—we found no obvious differences in their distribution patterns in various stages of meiotic prophase, indicating that the prophase I process was successfully completed in WT and *Ssh2* KO mice (*Figure 2—figure supplement 1*).

We performed detailed histological analysis by hematoxylin staining of testicular sections from WT and *Ssh2* KO males at various developmental stages. Testes collected from *Ssh2* KO mice at postnatal day (PD)7 and PD14 were indistinguishable from those of WT mice (*Figure 2—figure supplement*

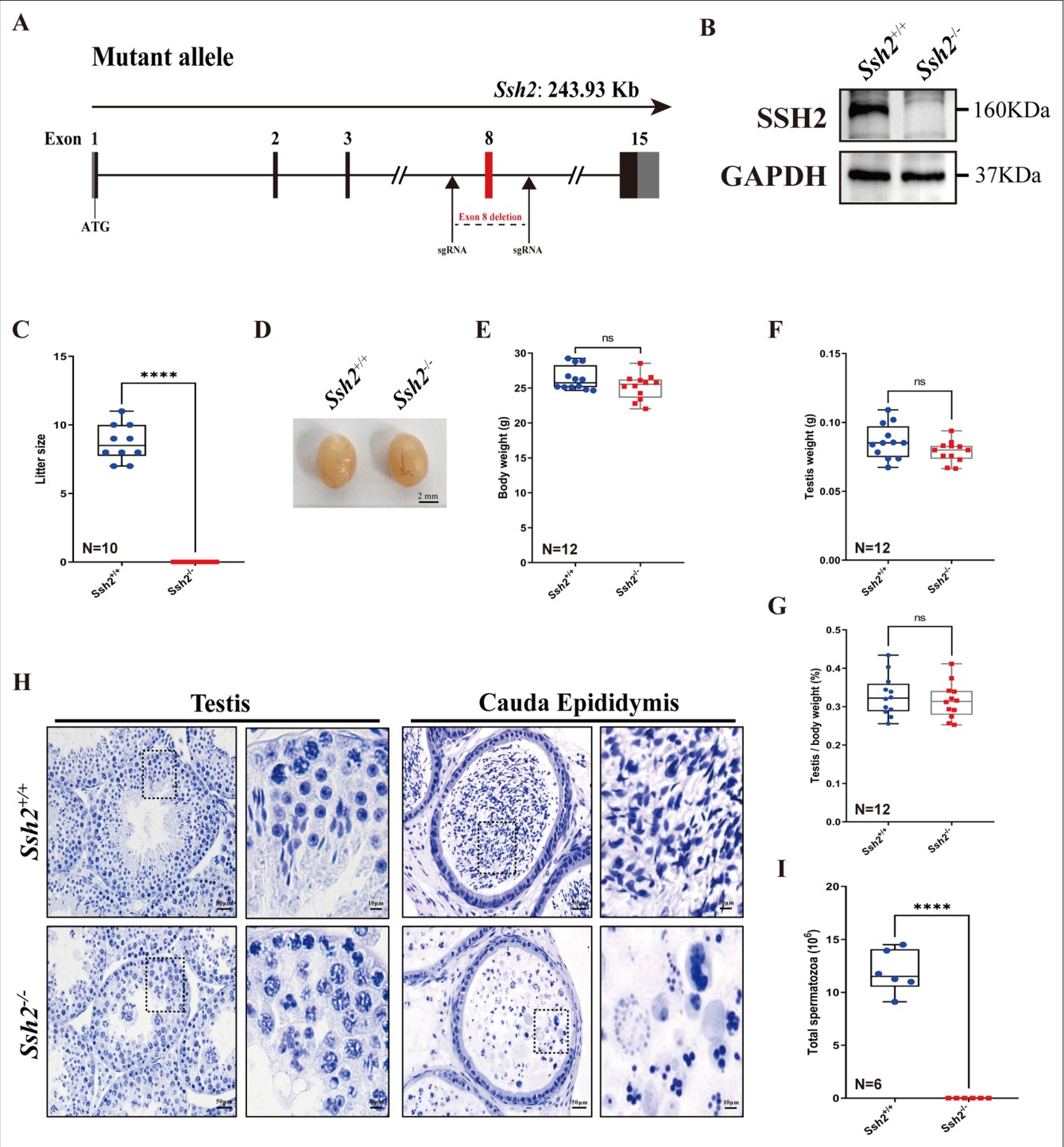

**Figure 1.** *Ssh2* knockout (KO) causes severe reproductive defects and male infertility in mice. (**A**) Schematic representation of the generation of *Ssh2* KO mice using CRISPR/Cas9. (**B**) Validation of *Ssh2* KO by western blotting in testicular lysates from wild-type (WT) and *Ssh2* KO 8-week-old mice (n=3), indicating the absence of SSH2 protein in *Ssh2* KO testes. GAPDH was used as the loading control. (**C**) Number of pups per litter from WT (8.70±0.42) and *Ssh2* KO (0.00) male mice (8 weeks of age) after crossing with WT female mice (8–10 weeks of age) for 3 months (n=10). Data are presented as the mean ± SEM; ****p<0.0001, calculated by Student's t-test. Bars indicate the range of data. (**D**) The testes from *Ssh2* KO mice appeared phenotypically normal when compared to testes of WT mice at 8 weeks of age, n=6. Scale bars: 2 mm. (**E–G**) Body weights (26.3609±0.4914 for WT; 25.1741±0.5189 for *Ssh2* KO), weights of the testes (0.0862±0.0036 for WT; 0.0788±0.0023 for *Ssh2* KO), and the testis-to-body weight ratio (0.3281±0.0153 for WT; 0.3154±0.0135 for *Ssh2* KO) of adult WT and *Ssh2* KO males which aged over 8 weeks (n=12). Data are presented as the mean ± SEM; p>0.05 calculated by Student's t-test. Bars indicate the range of data. (**H**) Histology of the testis and cauda epididymis from WT and *Ssh2* KO mice. Sections were stained

*Figure 1 continued on next page*

*Figure 1 continued*

with hematoxylin. No elongating or elongated spermatids were detected in *Ssh2* KO testes, and no mature sperm were detected in the *Ssh2* KO epididymis. Boxed regions are magnified on the right. Scale bars: 50 μm for the original region (left columns); 10 μm for the magnified region. Images are representative of testes/cauda epididymis extracted from at least six adult male mice per genotype. (I) Total sperm number in the cauda epididymis of WT (11.92×106 ± 0.82×106) and *Ssh2* KO (0.00) mice, n=6. Data are presented as the mean ± SEM; ****p<0.0001, calculated by Student's t-test. Bars indicate the range of data.

The online version of this article includes the following source data and figure supplement(s) for figure 1:

**Source data 1.** Observational datasets and original blots.

**Figure supplement 1.** PCR genotyping for wild-type (WT) and *Ssh2* knockout (KO) mice.

**Figure supplement 1—source data 1.** Original gels.

**Figure supplement 2.** Fluorescence-activated cell sorting (FACS) assessment in spermatogenic cells from wild-type (WT) and *Ssh2* knockout (KO) murine testes.

**Figure supplement 2—source data 1.** Original images.

*2*). The most advanced cells in WT testes were elongating spermatids at PD28, and none of these were present in *Ssh2* KO testes. Round spermatid clusters were first observed in *Ssh2* KO tubules at PD28, and these accumulated from PD35 to PD60. We next examined PD35 testes and histological images that indicated the *Ssh2* KO tubules were devoid of elongated spermatids (***Figure 2—figure supplement 2***).

To determine the developmental defects at specific stages of spermatogenesis, tissue sections of WT and *Ssh2* KO testes were stained with hematoxylin and periodic acid Schiff (PAS) and then labeled with acrosomal glycoproteins. We found that spermatogenesis of *Ssh2* KO mice was arrested at stages Ⅱ-Ⅲ (***Figure 2A***). Spermatids at various steps of spermiogenesis were observed in WT seminiferous tubules, whereas only step 1–3 round spermatids were present in *Ssh2* KO tubules, indicating spermiogenic failure in these mice. We also detected round spermatids with malformed acrosomal structures in *Ssh2* KO testes, which suggests that the acrosomal formation was also affected. Furthermore, clustered round spermatids and apoptotic-like germ cells were also observed in *Ssh2* KO mice tubules exhibiting small hyperchromatic nuclei and polynuclear structures, which are known indicator of apoptosis (***Catalano-Iniesta et al., 2019***; ***Figure 2A***).

We performed a terminal deoxynucleotidyl transferase-dUTP nick-end labeling (TUNEL) assay to identify the apoptotic cells in the testes and epididymides of WT and *Ssh2* KO mice. The counts of TUNEL-positive cells were significantly increased in the *Ssh2* KO seminiferous tubules which were observed primarily as apoptotic spermatids, as indicated by the nuclear morphology and location within spermatogenic epithelium (***Figure 2B***, ***Figure 2C***). Similarly, we found large numbers of isolated apoptotic spermatids in the lumen of epididymal ducts of *Ssh2* KO mice (***Figure 2B***, ***Figure 2D***). To understand the potential mechanism behind the enhanced germ cell apoptosis elicited by *Ssh2* KO, we measured the expression levels of several apoptotic signaling proteins including Caspase-3 with B-cell lymphoma-2 (BCL-2), BCL2-associated X protein (BAX), and BCL2-associated agonist of cell death (BAD), which are well known as Bcl-2 family members involved in Caspase-3 activation (***Gu et al., 2021***). As demonstrated by WB, in comparison with that of WT mice, remarkably elevated protein levels of pro-apoptotic BAX, BAD, Caspase-3, and cleaved Caspase-3 accompanied by relatively decreased protein levels of anti-apoptotic BCL-2 proteins were detected in the testicular lysates of *Ssh2* KO mice (***Figure 2E***), suggesting the induction of Caspase-3 activation in the spermatogenic cell apoptosis resulted from *Ssh2* KO. Consistently, these findings support the hypothesis that *Ssh2* KO resultant round spermatid arrest with malformed acrosomes during spermatogenesis and that ultimately causes spermatid apoptosis via the Bcl-2/Caspase-3 signaling pathway.

## *Ssh2* KO leads to disrupted acrosome biogenesis

To better understand the defects in *Ssh2* KO spermatid development, we performed an immunofluorescence analysis to evaluate the morphology of the differentiated spermatids in WT and *Ssh2* KO mice. Spermatids in distinct phases of acrosome biogenesis were stained with fluorescein-conjugated peanut agglutinin (PNA), a protein that specifically binds to the outer acrosomal membrane (***Cheng et al., 1996***). We found spermatids of all spermiogenic steps in the testes of WT males: as spermiogenesis progressed, singular acrosomal vesicles derived from proacrosomic vesicles were formed

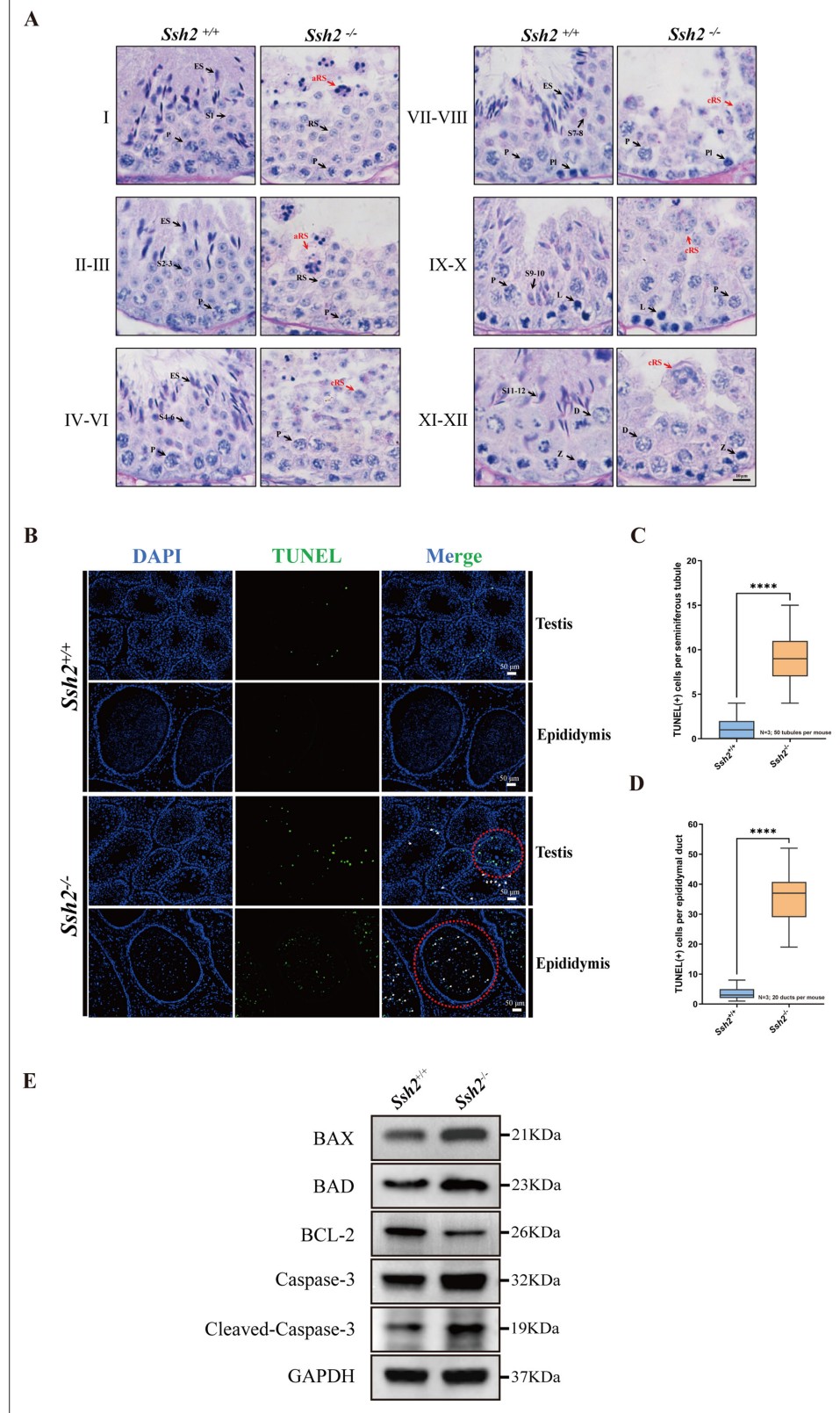

**Figure 2.** Spermatogenic arrest and enhanced germ cell apoptosis result from *Ssh2* knockout (KO). (**A**) Periodic acid Schiff (PAS)-hematoxylin-stained sections of seminiferous epithelia from wild-type (WT) mice and *Ssh2* KO mice (8 weeks of age, n=3), indicating spermatogenic arrest at steps 2–3 of spermatogenesis in the *Ssh2* KO mice. L, leptotene spermatocytes; D, diplotene spermatocytes; Z, zygotene spermatocytes; P, pachytene spermatocytes;

*Figure 2 continued on next page*

*Figure 2 continued*

RS, round spermatids; ES, elongated spermatids; S1–S12, spermatids at different spermiogenic steps; aRS: apoptotic-like round spermatids; cRS: clustered round spermatids. Scale bar: 10 µm. (**B**) Terminal deoxynucleotidyl transferase-dUTP nick-end labeling (TUNEL) immunofluorescence staining of the testicular and epididymal sections from WT and *Ssh2* KO mice. The TUNEL-positive puncta and signal intensity significantly increased in *Ssh2* KO testes and epididymides (as the red circles indicate). Green: TUNEL-positive signal; blue: DAPI; white arrows: TUNEL-positive germ cells. Scale bars: 50 µm. At least three mice (6–8 weeks of age) of each genotype were used in the analysis. (**C**) Counts of TUNEL-positive cells per seminiferous tubule in adult *Ssh2* KO testes (8.92±0.21) compared with control (1.19±0.08). Three mice of each genotype were assessed and 50 tubules were validated for each mouse. Data are shown as mean ± SEM; ****p<0.0001, calculated by Student's t-test. Bars indicate the range of data. (**D**) Counts of TUNEL-positive cells per epididymal duct in adult *Ssh2* KO testes (35.85±1.06) compared with control (3.57±0.21). Three mice of each genotype were assessed and 20 ducts were validated for each mouse. Data are shown as mean ± SEM; ****p<0.0001, calculated by Student's t-test. Bars indicate the range of data. (**E**) Immunoblotting analysis of BCL2-associated X protein (BAX), BCL2-associated agonist of cell death (BAD), B-cell lymphoma-2 (BCL-2), Caspase-3, and cleaved Caspase-3 in testicular lysates from adult WT and *Ssh2* KO mice, n=3; GAPDH was used as the loading control.

The online version of this article includes the following source data and figure supplement(s) for figure 2:

**Source data 1.** Terminal deoxynucleotidyl transferase-dUTP nick-end labeling (TUNEL)-related observational datasets and original images/blots.

**Figure supplement 1.** SSH2 is not required for meiotic progression in mouse spermatogenesis.

**Figure supplement 1—source data 1.** Original images.

**Figure supplement 2.** Comparison of spermatogenesis in wild-type (WT) and *Ssh2* knockout (KO) mice.

**Figure supplement 2—source data 1.** Original images.

in the Golgi phase (steps 1–3), became flattened during the cap phase (steps 4–6), spread to the laminae (steps 7–8), and were coated onto the nuclei in the acrosome/maturation phase (steps 9–16) (*Figure 3A*, upper panel). On the contrary, instead of intact acrosomal vesicles as seen in WT spermatids, multiple scattered proacrosomic vesicles separated from nuclei (without fusing with each other) were present in a majority of the *Ssh2* KO spermatids (steps 1–3) (*Figure 3A*, lower panel), demonstrating that acrosome biogenesis was disrupted starting from the Golgi phase in *Ssh2* KO mice.

## Proacrosomal vesicles fusion impairment in *Ssh2* KO spermatids

To gain insights into the defective acrosome formation in spermatids of *Ssh2* KO mice in the Golgi phase, which is characterized by the fusion of Golgi-derived proacrosomal vesicles (*Berruti and Paiardi, 2015*; *Khawar et al., 2019*), we next performed a transmission electron microscopy (TEM) analysis to explore the ultrastructure of the acrosome in spermatids in WT and *Ssh2* KO testes. TEM showed that WT spermatids had a single, large acrosomal vesicle in which the granule was attached to the concave surface of the nucleus and the inner acrosomal membrane was docked to the electron-dense acroplaxome. However, in *Ssh2* KO spermatids, multiple Golgi-derived small proacrosomal vesicles were adjacent to the trans-face of the Golgi apparatus which contain thick Golgi stacks and failed to fuse together to yield the acrosomal vesicle. Moreover, no acroplaxome-like structure was observed near the nuclear envelope (*Figure 3B*). These results suggest that the impairment of acrosomal formation in *Ssh2* KO mice might be due to the failure of proacrosomal vesicle fusion and/or defects in vesicular trafficking toward the nuclear surface.

Notably, such impaired acrosome biogenesis phenotype appears similar to observations in mice with mutations in *Gm130* (*Han et al., 2017*), which is known to function in Golgi membrane dynamics and fusion of Golgi-derived vesicles (*Koreishi et al., 2013*; *Walker et al., 2004*). To determine whether *Ssh2* KO affects the fusion of proacrosomal vesicles in the Golgi phase, we performed co-immunostaining of GM130 with PNA staining of seminiferous tubule squashes of WT and *Ssh2* KO mice. We observed GM130 signals with vesicle-like and stack-like shapes in close contact with the PNA-labeled acrosomal vesicles in WT spermatids. However, in *Ssh2* KO spermatids, only large GM130-positive fluorescent aggregates that exhibited no association with PNA-positive proacrosomal vesicles were observed (*Figure 3C*), suggesting that the localization of vesicular fusion-associated GM130 is altered in Golgi-phase spermatids by *Ssh2* KO. Collectively, these data indicate that *Ssh2* KO causes failure of proacrosomal vesicle fusion.

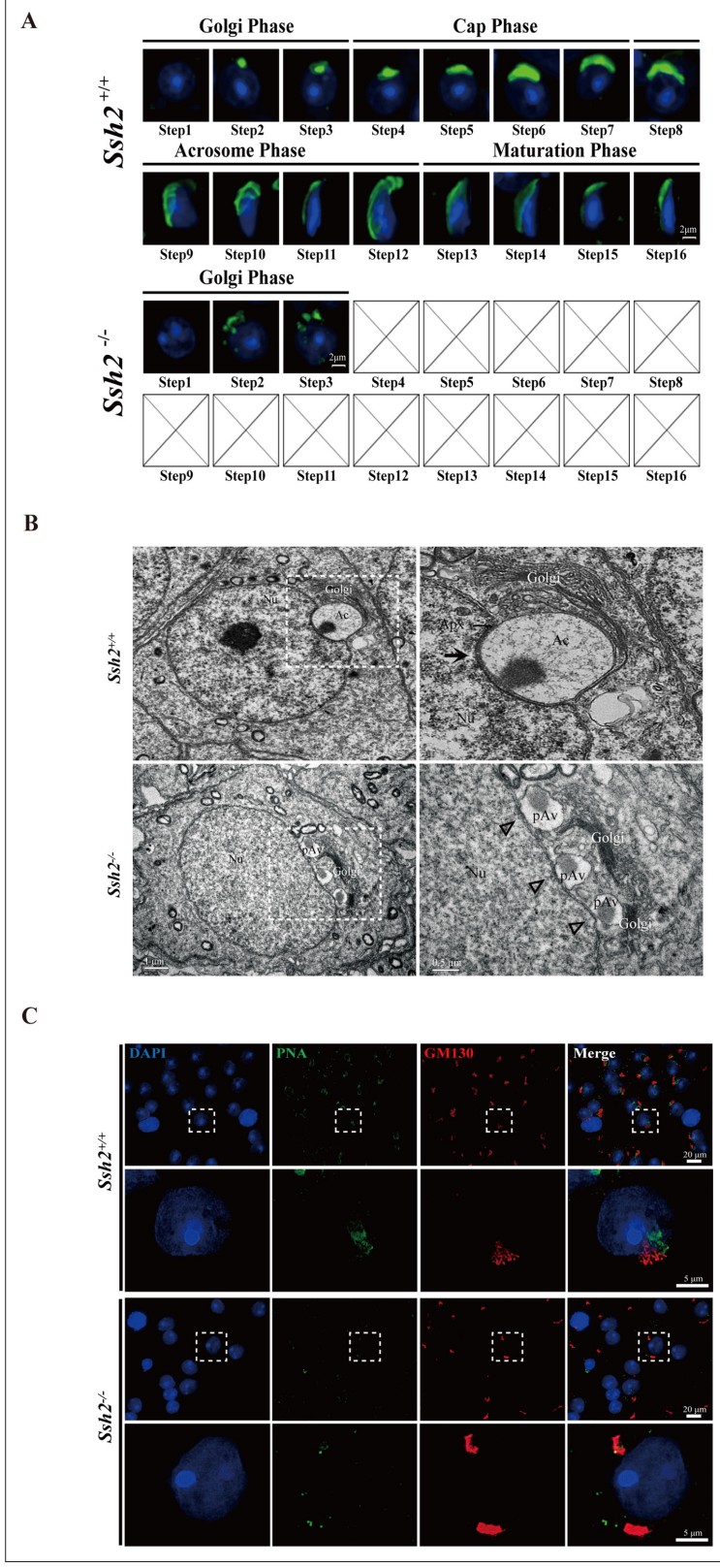

**Figure 3.** Acrosome biogenesis is disrupted during spermiogenesis in *Ssh2* knockout (KO) mice. (**A**) Analysis of spermiogenesis and acrosome biogenesis in wild-type (WT) and *Ssh2* KO mice (8 weeks of age, n=3) by fluorescence imaging of spermatids labeled with peanut agglutinin (PNA) lectin (green). Nuclei were stained with DAPI (blue). The phases of acrosome biogenesis and corresponding spermiogenesis steps are the Golgi phase

*Figure 3 continued on next page*

*Figure 3 continued*
(steps 1–3), cap phase (steps 4–7), acrosome phase (steps 8–12), and maturation phase (steps 13–16). Scale bars: 2 μm. (**B**) Ultrastructural analysis of WT and *Ssh2* KO spermatids. Intact acrosomes (black arrow) were observed in WT mice. The hollow triangles indicate proacrosomal vesicles that failed to fuse in *Ssh2* KO mice. The regions outlined by white boxes are shown at higher magnification to the right. Nu, nucleus; Ac, acrosome; Golgi, Golgi apparatus; Apx, acroplaxome; Pav, proacrosomal vesicle. Scale bars: left panel, 1 μm; right panel, 0.5 μm. (**C**) Immunofluorescence analysis of the vesicular fusion-related Golgi-specific protein GM130 (red) in WT and *Ssh2* KO round spermatids in testicular sections co-stained with PNA lectin (green). Nuclei were stained with DAPI (blue). Scale bars: original images, 20 μm; magnified images, 5 μm.

The online version of this article includes the following source data for figure 3:

**Source data 1.** Original images.

## SSH2 accumulates at the acrosomal region in round spermatids

To investigate the functional activity of SSH2 in spermatogenesis, we examined the pattern of SSH2 accumulation in testes at several developmental time points using WB. We observed a gradual increase in accumulation during murine spermatogenesis. The protein was undetectable in PD7 testes, became detectable at PD14, and substantially the level was increased by PD21, the time when round spermatids first appear in seminiferous tubules (*Guan et al., 2020*), suggesting a potential function of SSH2 in the development of spermatid (*Figure 4A*).

To determine the developmental roles of SSH2 in spermatogenic cells, we performed counterstaining of SSH2 with PNA on seminiferous tubule paraffin sections of WT and *Ssh2* KO mice. SSH2 accumulated predominantly in spermatocytes and post-meiotic round spermatids. In addition to the ubiquitous distribution observed in the cytoplasm of the germ cells, we noted bright, isolated SSH2-positive puncta positioned in the acrosomal region next to the nuclei of round spermatids (*Figure 4B*, left column). In contrast, fragmented acrosomes with a lack of SSH2 signal were detected in *Ssh2* KO spermatids (*Figure 4B*, right column).

To further understand the involvement of SSH2 in acrosome biogenesis, we assessed its subcellular localization in spermatids at different phases. Spermatids in seminiferous tubule squashes obtained from adult WT and *Ssh2* KO mice were immunostained with PNA and SSH2 for further observation in greater detail. Well-formed acrosomes in developing spermatids of WT mice exhibited diverse shapes at the different phases of acrosome biogenesis (*Figure 4C*, upper panels). In contrast, many irregulars, fractured PNA fluorescence signals with no cap-like PNA-positive structures were detected in *Ssh2* KO spermatids (*Figure 4C*, lower panels). Moreover, the association of SSH2 with the growing acrosome was evident by the acrosomal localization of SSH2 in the spermatids of WT mice (*Figure 4C*, upper panels) which was further confirmed by the distribution of HA-tagged SSH2 (SSH2-HA) detected on the testicular sections of the CRISPR/Cas9-generated viable mice harboring an HA-epitope fusion variant of SSH2. Immunofluorescence staining against the HA-tag showed that SSH2-HA is predominantly localized in the cytoplasm of spermatocytes and round spermatids with an intensive accumulation at the acrosomal region, consistent with the observations of the endogenous SSH2 (*Figure 4D*). Notably, punctate PNA signals which represent the nascent acrosome surrounded with robust SSH2-HA fluorescence signals were detected in Golgi-phase round spermatids within stage Ⅰ-Ⅲ tubules, instead of prevalent acrosomal localization of SSH2-HA in round spermatids at the cap phase of acrosome biogenesis, implying diverse roles of SSH2 in this continuous process (*Figure 4E*). Together, these results show that SSH2 is functionally involved in acrosome biogenesis.

### *Ssh2* KO alters F-actin organization in developing spermatids

It has been reported that SSH2 binds to F-actin in cultured HeLa cells, where it antagonizes LIMK1-induced actin polymerization (*Ohta et al., 2003*). To test whether F-actin organization in spermatids is affected by *Ssh2* KO during acrosome biogenesis, we first visualized F-actin structures in WT and *Ssh2* KO testicular sections at PD21, PD35, and PD60 using fluorochrome-conjugated phalloidin. As the representative images showed, the F-actin staining was altered in *Ssh2* KO spermatids. Unlike the expected sharp, sickle-shape heads of elongated spermatids surrounded by organized F-actin bundles seen in WT mice, the F-actin aggregated to form numerous lumps in *Ssh2* KO round spermatids (*Figure 5—figure supplement 1*). This aberrant accumulation of F-actin suggests that loss of *Ssh2* accelerates filament growth in spermatids rather than inducing filament disassembly.

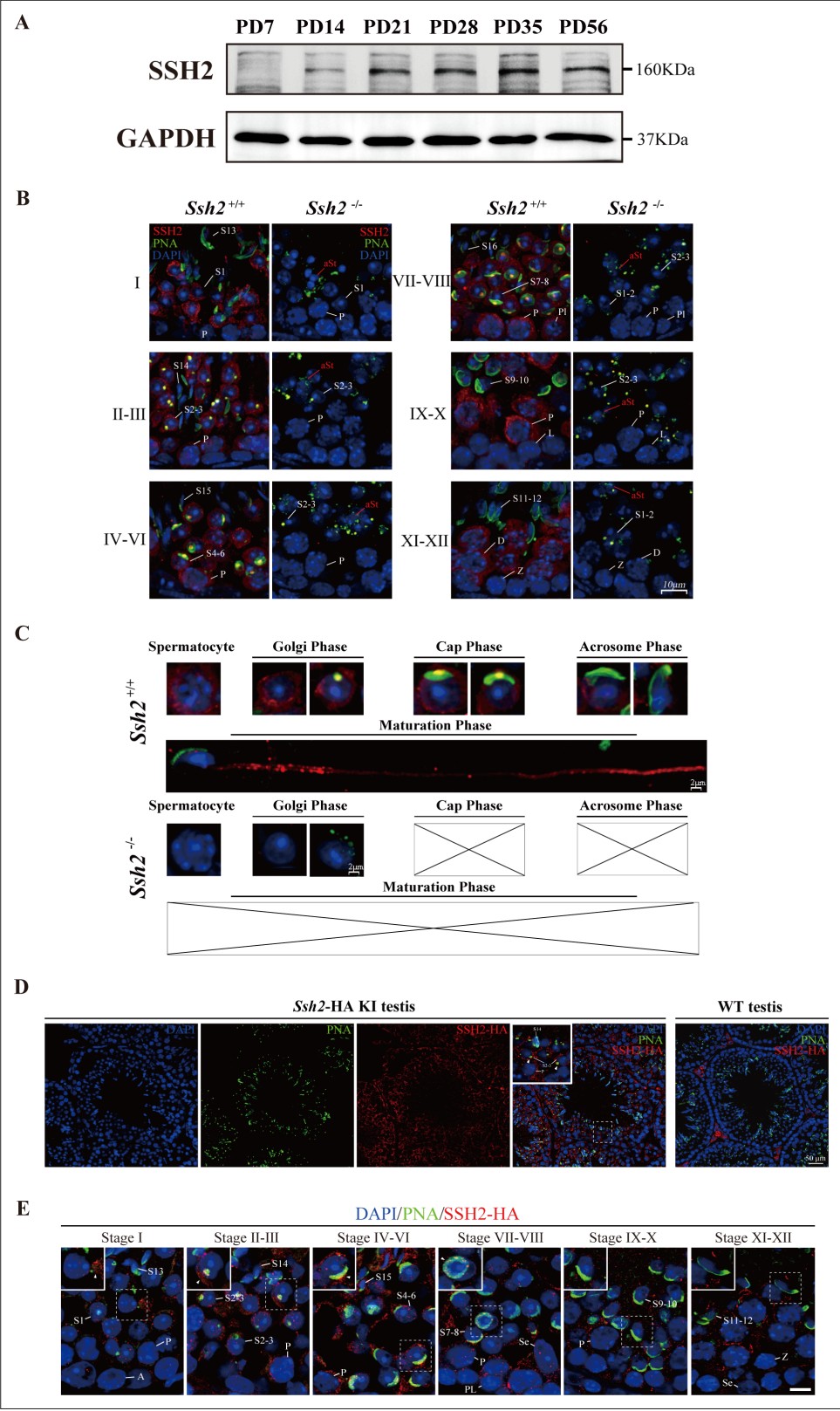

**Figure 4.** SSH2 accumulates at the acrosomal region of spermatids. (**A**) Immunoblotting against SSH2 in wild-type murine testes sampled at the indicated postnatal days (PD). The expression of SSH2 at PD14, PD21, PD28, PD35, and PD56 was measured. GAPDH was used as the loading control. (**B**) Co-immunofluorescence staining of Alexa Fluor 488-conjugated peanut agglutinin (PNA) lectin (green) and SSH2 (red) on testicular sections from

*Figure 4 continued on next page*

*Figure 4 continued*

wild-type (WT) (left lane) and *Ssh2* knockout (KO) (right lane) mice. Nuclei were stained with DAPI (blue). The epithelial spermatogenic cycle is routinely divided into 12 stages on the basis of changes in the morphology of the acrosome and nucleus in spermatids, and this was determined using the combination of PNA lectin and DAPI staining according to the established criteria. Cytoplasmic SSH2 localization in spermatocytes and spermatids of WT mice was observed, while fractured acrosomes were observed at all stages of spermatogenesis in *Ssh2* KO mice. P, pachytene spermatocytes; Pl, preleptotene spermatocytes; L, leptotene spermatocytes; Z, zygotene spermatocytes; D, diplotene spermatocytes; S1–16, step 1–16 spermatids. aRS: apoptotic-like spermatids. Scale bar: 10 µm. (**C**) Analysis of acrosomal morphogenesis in spermatogenic cells by co-staining of PNA lectin (green) and SSH2 (red) in WT (upper panel) and *Ssh2* KO (lower panel) murine testes. Nuclei were stained with DAPI (blue). Acrosomal morphology during acrosome biogenesis (Golgi, cap, acrosome, and maturation phase) is shown. No cap, acrosome, or maturation-phase spermatids were observed in *Ssh2* KO mice. Scale bar: 2 µm. (**D**) Immunofluorescence co-staining hemagglutinin (HA)-tag (red) with Alexa Fluor 488-conjugated PNA lectin (green) showing the predominant cytoplasmic SSH2 localization in spermatocytes and spermatids with an accumulation of SSH2 in acrosomal region in round spermatids on testicular sections of HA-tagged *Ssh2* KI mice while no obvious HA-tag fluorescence signal was detected in WT testes. Nuclei were stained with DAPI (blue). The framed regions are magnified. White triangles indicate the SSH2 acrosomal localization. P, pachytene spermatocytes; S2–3, step 2–3 spermatids; S14, step 14 spermatids. Scale bar: 50 µm. (**E**) Immunostaining of HA-tagged SSH2 (red) and PNA lectin (green) in testes of HA-tagged *Ssh2* knock-in (KI) mice indicating the dynamic subcellular localization of SSH2 in spermatids during the process of acrosome biogenesis. Nuclei were stained with DAPI (blue). Framed region containing representative spermatids at different phases of acrosome biogenesis are magnified. White triangles indicate the SSH2 acrosomal localization. A, type A spermatogonia; P, pachytene spermatocytes; Pl, preleptotene spermatocytes; Z, zygotene spermatocytes; S1–16, step 1–16 spermatids; Se, Sertoli cells. Scale bar: 10 µm.

The online version of this article includes the following source data for figure 4:

**Source data 1.** Original images/blots.

To further compare the successive changes in F-actin morphology during acrosome biogenesis in WT and *Ssh2* KO developing spermatids, we performed immunofluorescence microscopy on testicular sections co-stained with phalloidin and PNA. Phalloidin staining was diffusely distributed in the cytoplasm of round spermatids and intact F-actin bundles that were located around the laminal acrosomes in elongated spermatids of WT mice. In contrast, punctate phalloidin signals accompanied by thick F-actin bundles were mostly dissociated from the fractured acrosomes in *Ssh2* KO spermatids (*Figure 5A*). Moreover, we co-immunostained the spermatids with phalloidin and PNA in seminiferous tubule squashes from WT and *Ssh2* KO mice. As expected, confocal microscopy showed that punctate F-actin signals were distributed ubiquitously throughout the cytoplasm of Golgi-phase spermatids of WT mice, with relatively intense fluorescence signals localized at acrosomal regions. However, robust phalloidin staining was scattered irregularly in *Ssh2* KO spermatids and exhibited no association with the acrosomal signals (*Figure 5B*). Together, these findings indicate that F-actin is highly disorganized in *Ssh2* KO spermatids during acrosome biogenesis.

### *Ssh2* KO impairs proacrosomal vesicle trafficking

Given the disrupted acrosome biogenesis and disorganized F-actin in *Ssh2* KO spermatids, we hypothesized that SSH2 facilitates Golgi-acrosome vesicular trafficking by modulating F-actin organization. To test this hypothesis, we performed immunostaining against PNA and GOPC (an acrosome-related protein implicated in vesicle trafficking from the Golgi apparatus to the acrosome; *Yao et al., 2002*) in WT and *Ssh2* KO spermatids in seminiferous tubule squashes. In WT Golgi-phase spermatids, GOPC retained its distinct distribution, which was predominantly confined to the acrosomal region as indicated by its colocalization with PNA-positive structures near the nuclei (76.50 ± 3.70%). In contrast, in *Ssh2* KO spermatids GOPC exhibited dispersed localization in the cytoplasm and the extent of GOPC-PNA colocalization was significantly diminished (23.75 ± 7.72%) (*Figure 6A and B*). These results suggest that the participation of GOPC in proacrosomal vesicle trafficking is negatively affected in *Ssh2* KO mice.

We then tested whether the localization of the autophagy-related protein LC3, which is reported to function in Golgi-derived proacrosomal vesicle trafficking through its membrane conjugation (*Wang et al., 2014b*), was affected by *Ssh2* KO. Immunofluorescence analysis of LC3 was conducted on seminiferous tubule squashes from WT and *Ssh2* KO mice. We observed that LC3 was localized on

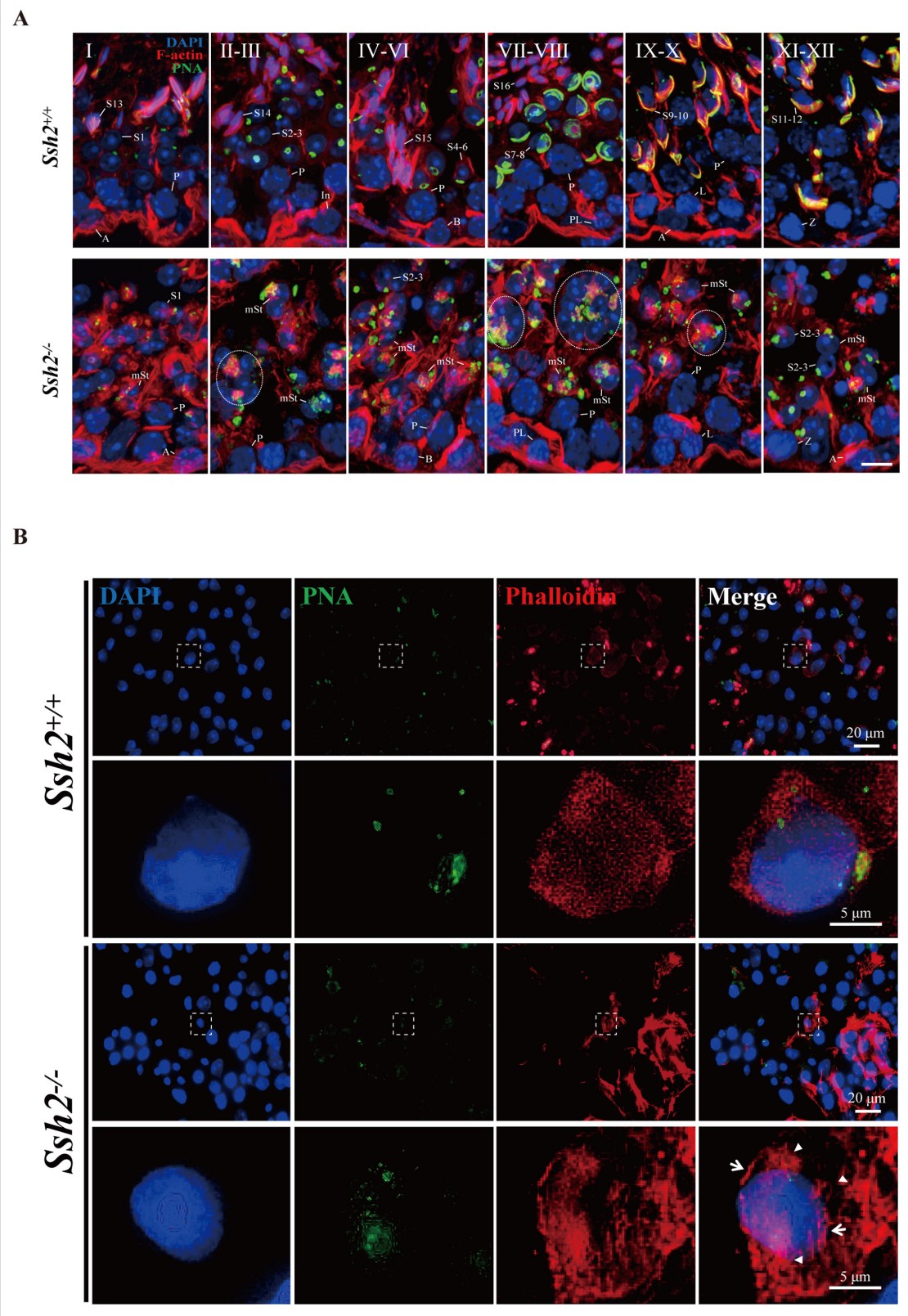

**Figure 5.** *Ssh2* knockout (KO) results in disturbed filamentous actin (F-actin) remodeling in spermatids. (**A**) Immunofluorescence detection of F-actin (red) and peanut agglutinin (PNA) lectin (green) during spermatogenesis in testicular sections from wild-type (WT) (top lane) and *Ssh2* KO (bottom lane) mice at postnatal day (PD)85. Blocky actin filaments (as indicated by the circles) were observed in spermatids with malformed acrosomes. A, type A spermatogonia; In, intermediate spermatogonia; B, type B spermatogonia; P, pachytene spermatocytes; Pl, preleptotene spermatocytes; L, leptotene

*Figure 5 continued on next page*

*Figure 5 continued*

spermatocytes; Z, zygotene spermatocytes; D, diplotene spermatocytes; S1–16, step 1–16 spermatids; mSt, spermatids with malformed acrosomes. Scale bar: 10 µm. (**B**) Fluorescence analysis of phalloidin (red) and PNA lectin (green) on tubule squashes from adult WT and *Ssh2* KO mice. Nuclei were stained with DAPI (blue). Diminutive actin filaments exhibited a uniform cytoplasmic distribution in WT spermatids, whereas the actin remodeling was disrupted as thick actin filaments (indicated by white arrows) and lumpy actin aggregates (indicated by white triangles) were formed in *Ssh2* KO spermatids indicated by phalloidin staining. The framed regions are magnified beneath. Scale bars: original images, 20 µm; magnified images, 5 µm.

The online version of this article includes the following source data and figure supplement(s) for figure 5:

**Source data 1.** Original images.

**Figure supplement 1.** Disordered filamentous actin (F-actin) in *Ssh2* knockout (KO) testes.

**Figure supplement 1—source data 1.** Original images.

proacrosomal vesicles in most of the WT spermatids (89.45 ± 2.50%), as indicated by its colocalization with PNA. In contrast, vesicle-like LC3-positive fluorescent signals surrounded the PNA-positive structures, and a portion of the LC3 molecules was not recruited to proacrosome vesicles in *Ssh2* KO spermatids (77.75 ± 5.91%) (**Figure 6C and D**). Together, these results indicate that *Ssh2* KO results in impaired trafficking of proacrosomal vesicles to the growing acrosome.

### *Ssh2* KO spermatids display impaired COFILIN phospho-regulation and thus have disturbed F-actin remodeling

In considering the potential molecular mechanisms underlying the observed impairment of acrosome biogenesis in *Ssh2* KO mice, we focused on the known role of SSH2 as a COFILIN phosphatase (**Ohta et al., 2003**). Given our observation of disorganized F-actin in *Ssh2* KO spermatids, we speculated that SSH2 acts as a modulator in actin remodeling through COFILIN dephosphorylation, which is essential for spermiogenesis. To pursue this further, we monitored the expression of COFILIN, phospho-COFILIN (p-COFILIN), and protein kinases that directly phosphorylate COFILIN (e.g., LIMK1 and LIMK2; **Arber et al., 1998**) by WB of testicular lysates of PD82 WT and *Ssh2* KO testes. The *Ssh2* KO mice had elevated levels of p-COFILIN compared to their WT littermates, whereas there were no significant differences in the levels of total COFILIN, LIMK1, or LIMK2 (**Figure 7A**, **Figure 7—figure supplement 1**). Accordingly, we also conducted an immunofluorescent analysis on testicular sections from adult WT and *Ssh2* KO mice using the antibody against p-COFILIN. Notwithstanding some mild nuclear staining in round spermatids, we did not observe evident p-COFILIN signals in the germ cells in WT testes. As expected, compared to WT mice, the signal intensity of p-COFILIN was dramatically increased in both the nucleus and cytoplasm of *Ssh2* KO spermatogenic cells, specifically in spermatids (**Figure 7B**). These findings suggest that COFILIN phospho-regulation is impaired in *Ssh2* KO testes with subsequent accumulation of p-COFILIN.

To further determine whether the effect of SSH2 on the actin cytoskeleton is mediated by the control of COFILIN phospho-regulation, we performed both WB and immunostaining against SSH2, p-COFILIN, and F-ACTIN in WT and *Ssh2* KO testicular lysates and sections, respectively. Not surprisingly, we observed increased expression of F-actin accompanied by p-COFILIN in *Ssh2* KO testes (**Figure 7C**). Fluorescence images also showed ubiquitous p-COFILIN signals overlapping with disorganized F-actin aggregates in *Ssh2* KO spermatids (**Figure 7D**). Collectively, these findings support a role for SSH2 in orchestrating F-actin remodeling through the control of COFILIN phosphorylation during acrosome biogenesis.

## Discussion

Although SSHs exert COFILIN phosphatase activity in mammals, the organ distribution of SSH family members varies significantly in mice, implying their similar but diverse biochemical functions (**Ohta et al., 2003**). In the present study, we show the essential role of SSH2 in acrosome biogenesis and male fertility based on the genetically engineered KO and HA-tagged KI mice targeting *Ssh2*. Our findings demonstrate that the reproductive phenotype of severe spermatogenic arrest at the early round spermatid stage with enhanced germ cell apoptosis leads to subsequent male infertility in *Ssh2* KO mice. The males lacking functional SSH2 display disrupted germ cell development starting from the spermiogenic step 2–3 round spermatids. Moreover, we characterize predominantly cytoplasmic

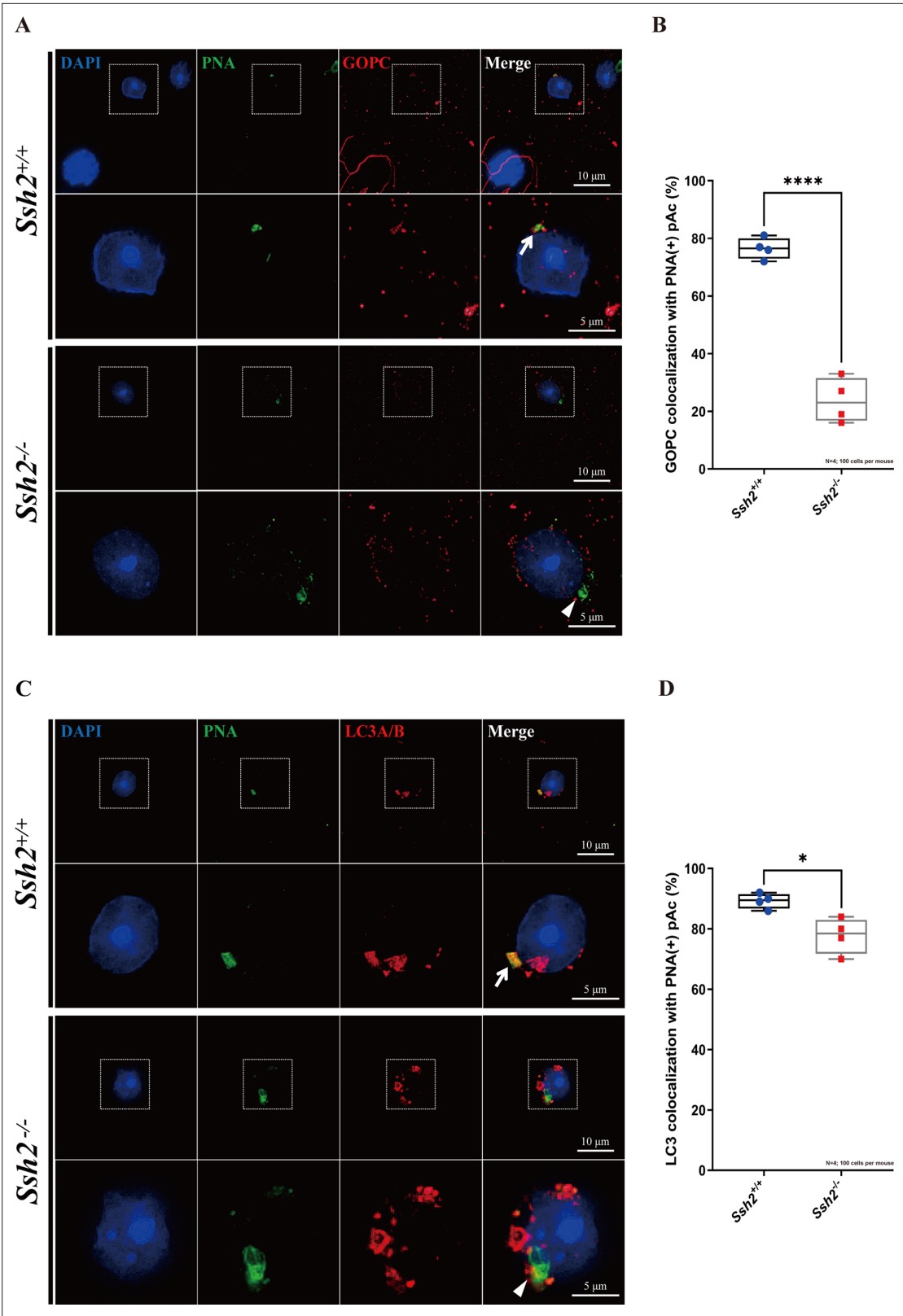

**Figure 6.** *Ssh2* knockout (KO) spermatids exhibit defects in proacrosomal vesicle transport. (**A**) Immunofluorescence staining of the vesicular trafficking-related Golgi-specific protein GOPC (red) in wild-type (WT) and *Ssh2* KO round spermatids in testicular sections co-stained with peanut agglutinin (PNA) lectin (green). Nuclei were stained with DAPI (blue). Acrosomal debris was observed in *Ssh2* KO round spermatids. GOPC colocalized with PNA lectin in WT spermatids (upper panels, arrow indicates the colocalization) but not in those of *Ssh2* KO mice (lower panels, triangle indicates the absence of

*Figure 6 continued on next page*

*Figure 6 continued*

colocalization), as shown in the representative images. Framed areas are magnified beneath. Scale bars: original images, 10 μm; magnified images, 5 μm. (**B**) Quantitative analysis of GOPC and PNA lectin colocalization in WT mice, 76.50 ± 3.70%; *Ssh2* KO mice, 23.75 ± 7.72% (n=4; 400 cells). Data are shown as mean ± SEM, ***p<0.001 by Student's t-test. Bars indicate the range of data. (**C**) Immunofluorescence staining of the autophagosome-related protein LC3A/B (red) in WT and *Ssh2* KO round spermatids on the testicular sections co-stained with PNA lectin (green). Nuclei were stained with DAPI (blue). LC3A/B colocalized with PNA lectin in WT spermatids (upper panels, arrow indicates the colocalization) but not in those of *Ssh2* KO mice (lower panels, triangle indicates the absence of colocalization) as representative images showed. Framed areas are magnified beneath. Scale bars: original images, 10 μm; magnified images, 5 μm. (**D**) Quantitative analysis of LC3A/B and PNA lectin colocalization in WT mice, 89.45 ± 2.50%; *Ssh2* KO mice, 77.75 ± 5.91% (n=4; 400 cells). Data are shown as the mean ± SEM, ***p<0.05 by Student's t-test. Bars indicate the range of data.

The online version of this article includes the following source data for figure 6:

**Source data 1.** Observational datasets and original images.

localization of SSH2 in post-meiotic round spermatids with accumulation at acrosomal region in testes of WT mice that was confirmed by the observation in HA-tagged *Ssh2* KI mice. Further examinations showed aberrant acrosome morphology in the *Ssh2* KO spermatids during spermiogenesis, characterized by fragmented acrosomal debris without the fusion of proacrosomal vesicles. We also observed disorganized F-actin in the mutant developing spermatids accompanied by impaired proacrosomal vesicle trafficking, and this suggests a fundamental role for F-actin organization in acrosome formation. In addition, enhanced phosphorylation of COFILIN with thick F-actin fibers was seen in *Ssh2* KO testes. We interpret these findings as evidence of a role for SSH2 in acrosome biogenesis via the regulation of COFILIN-mediated actin remodeling (*Figure 7E*).

The implication of COFILIN during reproductive developmental process has been described in model animals, as evidenced by the embryonic lethality of *Cofilin-1* KO mice (*Gurniak et al., 2005*). Non-muscle COFILIN was identified as a component of tubulobulbar complexes in rat Sertoli cells (*Guttman et al., 2004*) and as a key activator of human sperm capacitation and the acrosome reaction (*Megnagi et al., 2015*), but its functions in mammalian spermatogenesis have remained largely elusive. Our findings of increased apoptotic index in *Ssh2* KO testes are consistent with the data of abnormal testes phenotype in mice with *Limk2* KO (*Takahashi et al., 2002*), suggesting a potential role for COFILIN in regulating the germ cell cycle in mice. Another line of evidence linking COFILIN to apoptosis comes from the investigations performed in cell context. Treating A549 cells with thapsigargin, an inhibitor of endoplasmic reticulum calcium ATPase, also induces apoptosis through the inhibition of COFILIN-mediated F-actin reorganization (*Wang et al., 2014a*). Elevated COFILIN phosphorylation after SSH2 knockdown was previously demonstrated to induce the activation of Caspase-3/7 in a human renal cell carcinoma cell line (*Lu et al., 2014*), and activation of Caspase-3/7 is known to trigger apoptotic cell death (*Saller et al., 2010*). Consistently, we observed distinct Caspase-3 activation accompanied by correlative changes in the expression of Bcl-2 family members in *Ssh2* KO testes. Based on data from previous reports and comparing it with our findings, we speculate that the mutant spermatids in *Ssh2* KO mice possibly undergo cell cycle arrest via the Bcl-2/Caspase cascade due to impaired COFILIN phosphorylation. Future studies are needed to further elucidate the mechanisms underlying this actin-based process.

Owing to the resolution limitation of microscopy, we could not directly image the ultrastructure of F-actin tracts between the Golgi body and the acrosome. However, the observation from TEM that the lack of the electron-dense acroplaxome, a cytoskeletal actin-rich scaffold plate that anchors the acrosome to the nuclear envelope (*Kierszenbaum et al., 2003a*), in *Ssh2* KO spermatids indicates a potential role of SSH2 in the shaping of the acroplaxome. Notably, defective acroplaxome structures with impaired acrosome biogenesis were observed in mutant spermatids lacking homeodomain-interacting protein kinase 4 (*Crapster et al., 2020*), which is responsible for the regulation of F-actin remodeling during spermiogenesis. We infer from these findings that acrosome biogenesis in *Ssh2* KO mice might also be affected by the acroplaxome defects that result from disordered F-actin remodeling, and further experiments should be performed to confirm this.

Given the diverse physiological roles reported for Slingshot family proteins, the possibility of an alternative mechanism underlying the involvement of SSH2 in cellular events beyond the COFILIN-mediated actin remodeling should be noted. According to some publicly accessible databases as the indicators of potential protein-protein interactions such as BioGRID (*Oughtred et al., 2019*) and IntAct (*Del Toro et al., 2022*), SSH2 might interact with a set of actin-based molecular motors covering

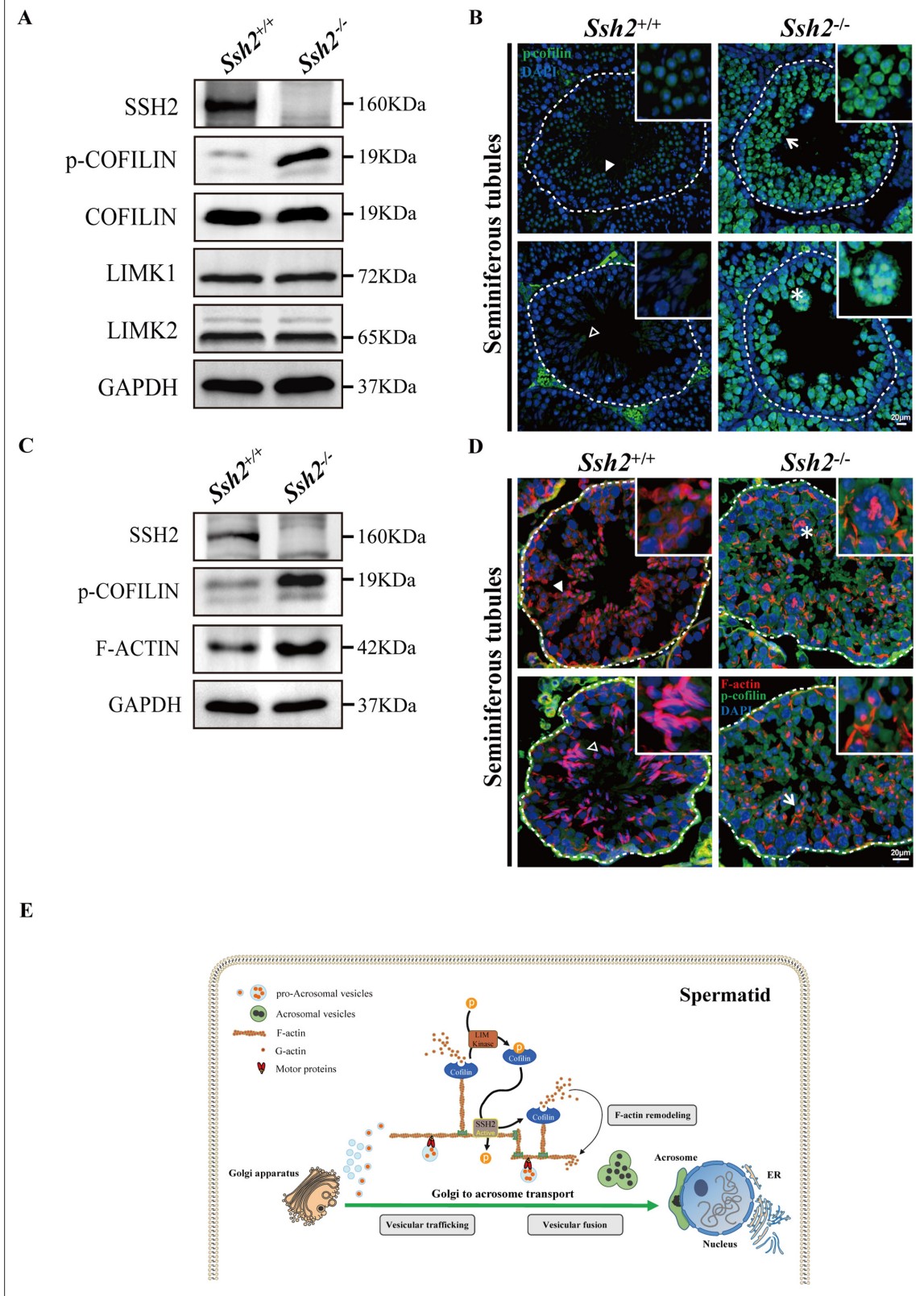

**Figure 7.** *Ssh2* knockout (KO) spermatids display impaired COFILIN phospho-regulation that disrupts filamentous actin (F-actin) remodeling. (**A**) Immunoblotting against SSH2, p-COFILIN, COFILIN, LIMK1, and LIMK2 in wild-type (WT) and *Ssh2* KO testes sampled on postnatal day (PD)82. GAPDH was used as the loading control. (**B**) Testicular sections of WT and *Ssh2* KO mice stained with p-COFILIN (green) and DAPI (blue), n=3. The solid and hollow triangles point to the round spermatids and elongated spermatids of low p-COFILIN expression in seminiferous tubules of WT mice. The white

*Figure 7 continued on next page*

*Figure 7 continued*

arrow points to the round spermatids of strong p-COFILIN expression in seminiferous tubules of *Ssh2* KO mice. The asterisk indicates a germ cell cluster. Scale bars: 20 µm. (**C**) Western blot analysis of SSH2, p-COFILIN, and F-ACTIN in lysates from 8-week-old WT and *Ssh2* KO testes, n=3; GAPDH was used as the loading control. (**D**) Testicular sections of 8-week-old WT and *Ssh2* KO mice stained for p-COFILIN (green) and F-actin (red), n=3. The solid and hollow triangles point to the round spermatids and elongated spermatids with intact F-actin organization in seminiferous tubules of WT mice. The white arrow points to the round spermatids of strong p-COFILIN expression with disorganized F-actin in seminiferous tubules of *Ssh2* KO mice. The asterisk indicates a germ cell cluster. Scale bars: 20 µm. (**E**) Proposed model for the functional role of the SSH2-COFILIN pathway in acrosome biogenesis. SSH2 participates in F-actin remodeling by regulating the phosphorylation of COFILIN. Intact and organized F-actin dynamics enables the transport of proacrosomal vesicles from the Golgi apparatus to the apical pole of the nucleus of the spermatid. F-actin also participates in the fusion of Golgi-derived vesicles and extra-Golgi vesicles.

The online version of this article includes the following source data and figure supplement(s) for figure 7:

**Source data 1.** Original images/blots.

**Figure supplement 1.** COFILIN-associated protein expression level in wild-type (WT) and *Ssh2* knockout (KO) testes.

**Figure supplement 1—source data 1.** Original blots.

MYH9, MYO19, and MYO18A, which have been implicated in the maintenance of Golgi morphology and Golgi anterograde vesicular trafficking via the PI4P/GOLPH3/MYO18A/F-actin pathway (*Rahajeng et al., 2019*). Moreover, it is intriguing to note that the phenotypes of *Ssh2* KO mice share a lot of similarities with that of *Pick1* KO model (*Xiao et al., 2009*) such as acrosome fragmentation and enhanced germ cell apoptosis, suggesting the possibility that SSH2 and PICK1 work together in the same trafficking machinery functioning in acrosome biogenesis which needs to be clarified further.

In conclusion, our study confirms the requirement for SSH2 during mammalian spermatogenesis, specifically in acrosomal biogenesis. The Slingshot family is evolutionarily conserved in mammals, therefore we believe that some mutations in the *SSH2* gene should exist in non-obstructive azoospermia, although no report was published yet that highlighting such mutation. The abnormal phenotype *Ssh2* KO in the mouse model may be valuable to understand the pathological factors causing azoospermia in humans.

## Materials and methods

### Animals

Animals used in this study were housed under controlled environmental conditions with 12 hr light cycles and free access to pathogen-free water and food. All animal-breeding work was carried out at the Laboratory Animal Center of Shandong University where the mice colony was maintained and samples were collected according to protocols approved by the Animal Ethics Committee of the School of Medicine, Shandong University. In this study, all animal care protocols were reviewed and approved by the Animal Use Committee of the School of Medicine, Shandong University.

### Generation of *Ssh2* KO mice

The mouse *Ssh2* gene with a length of 243.93 kb (Ensembl: ENSMUSG00000037926) has been identified that located on chromosomes 11, and 15 exons. To generate *Ssh2* KO mice in the C57BL/6J background, the CRISPR/Cas9 genome editing system (Cyagen Biosciences, Suzhou, China) was used. Briefly, single guide RNAs (sgRNAs) targeting the mouse *Ssh2* gene were co-injected with the Cas9 mRNA into fertilized mouse eggs, resulting in the deletion of the genomic DNA segment harboring exon 8 of *Ssh2*. Founder mice were identified by PCR followed by sequence analysis to confirm the deletion, and these mice were crossed with WT mice to test germline transmission and F1 animal generation.

### Generation of HA-tagged *Ssh2* KI mice

KI mice expressing HA-tagged SSH2 were generated and supplied by Genome Tagging Project (GTP) Center (Center for Excellence in Molecular Cell Science, CAS, Shanghai, China), using CRISPR/Cas9-mediated recombination targeting the C-terminal of *Ssh2*. Briefly, sgRNA (AAAATCACGTGTGTGGGCTC) was synthesized and ligated to the pX330-mCherry plasmid (#98750, Addgene, MA, USA) to construct the CRISPR-Cas9 plasmid. The sequences encoding the left homologous arm, HA tag, and

right homologous arm were amplified and ligated to the linear pMD19T vector with 20 bp overlap in order by Seamless Cloning Kit (D7010S, Beyotime, Beijing, China) to construct the HA tag KI DNA donor. DKO-AG-haESCs (*Zhong and Li, 2017*) were co-transfected with CRISPR-Cas9 plasmid and HA tag KI DNA donor, and the positive clones that possessed proper KI allele were selected by genomic DNA PCR amplification for Sanger sequencing. DKO-AG-haESCs of Ssh2-C-HA arrested at M phase were used for intracytoplasmic injection to generate semi-cloned embryos as described previously (*Zhong et al., 2015*) and 15–20 two-cell embryos were transferred into each oviduct of pseudopregnant ICR females. Generated chimeric female mice (F0) were mated with WT C57BL/6J male mice to obtain the HA-tagged *Ssh2* KI heterozygous mice which underwent subsequent mating and resulted in homozygous HA-tagged *Ssh2* KI mice were used for the study.

## Genotyping

Genomic DNA extracted from mouse tail tips (*Liu et al., 2019*) was subjected to PCR performed in a thermal cycler (T100, Bio-Rad Laboratories, Hercules, CA, USA). For genotyping of *Ssh2* KO mice, the WT and *Ssh2* KO alleles were assayed by primers: 5'-TCC TTG CCT TGA GAA TTC AAG CAA G-3' (forward) and 5'-TGA TAT GGT TAG TCC ATT GTG CCC A-3' (reverse). The corresponding PCR conditions were set as follows: 94°C for 3 min; 35 cycles of 94°C for 30 s, 66°C for 35 s, and 72°C for 35 s; and 72°C for 5 min. For the genotyping of HA-tagged *Ssh2* KI mice, the WT allele was assayed by primers: 5'-CCC TTC TAT AAC ACC ATG TGA TCC TG-3' (forward) and 5'-CCT TCA GCA GAT AGG TCG CC-3' (reverse); the KI allele was assayed by primers: 5'-AAA CTG GAC CCG TCA CCT G-3' (forward) and 5'-ATC CGG CAC ATC ATA CGG AT-3' (reverse). The corresponding PCR conditions were set as follows: 95°C for 7 min; 35 cycles of 95°C for 15 s, 60°C for 20 s, and 72°C for 45 s; and 72°C for 10 min. The PCR products were assessed by using 2% agarose gel.

## Antibodies and reagents

The rabbit anti-SSH2 polyclonal antibody against amino acids 1295–1423 of the mouse SSH2 protein was custom-generated by Dia-an Biological Technology Incorporation (Wuhan, China) as previously described (*Li et al., 2019*). The customized SSH2 antibody was used for WB with dilution (1:1000) and for immunofluorescent staining, the dilution was 1:200. The antibodies for WB were rabbit anti-COFILIN monoclonal antibody (1:1000 dilution, 5175, Cell Signaling Technology, Danvers, MA, USA), rabbit anti-p-COFILIN monoclonal antibody (1:1000 dilution, 3313, Cell Signaling Technology), mouse anti-LIMK-1 monoclonal antibody (1:100 dilution, sc-515585, Santa Cruz Biotechnology, Dallas, TX, USA), mouse anti-LIMK-2 monoclonal antibody (1:100 dilution, sc-365414, Santa Cruz Biotechnology), mouse anti-F-ACTIN monoclonal antibody (1:500 dilution, ab130935, Abcam), rabbit anti-BAX antibody (1:1000 dilution, 2772, Cell Signaling Technology), rabbit anti-BAD antibody (1:1000 dilution, 9292, Cell Signaling Technology), rabbit anti-Caspase-3 antibody (1:1000 dilution, 9662, Cell Signaling Technology), rabbit anti-Cleaved Caspase-3 monoclonal antibody (1:1000 dilution, 9664, Cell Signaling Technology), rabbit anti-BCL-2 polyclonal antibody (1:1000 dilution, 26593-1-AP, Proteintech, Rosemont, IL, USA), rabbit anti-HA tag polyclonal antibody (1:1000 dilution, 51064-2-AP, Proteintech), and mouse anti-GAPDH monoclonal antibody (1:10,000 dilution, 60004-1-lg, Proteintech). HRP-conjugated Affinipure goat anti-mouse and anti-rabbit IgG (H+L) (SA00001-1/SA00001-2, Proteintech) was used as the secondary antibodies for WB. The antibodies for immunofluorescent staining were mouse anti-p-COFILIN monoclonal antibody (1:100 dilution, sc-271921, Santa Cruz Biotechnology), mouse anti-GM130 monoclonal antibody (1:100 dilution, 610822, BD Biosciences, San Jose, CA, USA), rabbit anti-GOPC polyclonal antibody (1:100 dilution, ab37036, Abcam, Cambridge, UK), rabbit anti-LC3 polyclonal antibody (1:200 dilution, 4108, Cell Signaling Technology), rabbit anti-HA tag polyclonal antibody (1:100 dilution, 51064-2-AP, Proteintech), rabbit anti-SYCP1 polyclonal antibody (1:500 dilution, ab15090, Abcam), and mouse anti-SYCP3 monoclonal antibody (1:200 dilution, ab97672, Abcam). The secondary antibodies used for immunofluorescent staining were Alexa Fluor 488/594-conjugated goat anti-rabbit IgG (H+L) (A-11008/A-11012, Invitrogen, Carlsbad, CA, USA) and goat anti-mouse IgG H&L (Alexa Fluor 488/594) (ab150117/ab150120, Abcam). Rhodamine phalloidin was used to visualize F-actin (R415, Invitrogen), and Alexa Fluor 488-conjugated lectin PNA from *Arachis hypogaea* was used to visualize acrosomes (L21409, Invitrogen).

## In vivo fertility assessment

For the fertility test, adult male mice (n=3) of different genotypes with the age of 8–10 weeks were used to mate with 2–3 C57BL/6J female mice of the same age for 3 months. Pregnant females were confirmed by the presence of a vaginal plug. The number of live pups in every cage was counted individually.

## Tissue collection, histological analysis, and immunofluorescence

For histological examination, at least three-adult mice for each genotype were analyzed. Testes and caudal epididymides were dissected immediately following euthanasia, fixed in Bouin's solution (HT1013, Sigma-Aldrich, St Louis, MO, USA) for 16–24 hr at room temperature, dehydrated in an ethanol series (70%, 85%, 90%, 95%, and absolute ethanol), cleared in xylene, and then finally embedded in paraffin. For immunofluorescent staining, testes and epididymides samples were fixed in 4% paraformaldehyde (PFA, P1110, Solarbio, Beijing, China) overnight at 4°C, dehydrated, cleared, and embedded. The tissues were cut into 5 µm sections using a microtome (HistoCore BIOCUT, Leica Biosystems, Nussloch, Germany) and coated onto the glass slides. Following drying at 60°C for 1 hr, the sections were deparaffinized in xylene, hydrated by a graded alcohol series, and stained with hematoxylin for histological analysis or stained with PAS-hematoxylin (ab150680, Abcam) for determining the seminiferous epithelia cycle stages. For TUNEL staining, we followed the manufacturer's instructions (KeyGen Biotech, Nanjing, China). Images were collected under a microscope with a coupled camera device (BX53, Olympus, Tokyo, Japan). For immunostaining, deparaffinized sections were hydrated, immersed in sodium citrate buffer (pH 6.0), and heated for 15 min in a boiling water bath for antigen retrieval. After permeabilization with PBS containing 0.3% Triton X-100 and blocking with 5% bovine serum albumin, the slides were incubated with primary antibodies overnight at 4°C. Then the sections were rinsed in PBS and incubated with appropriate FITC-conjugated secondary antibodies and/or Alexa Fluor 488-conjugated PNA/rhodamine phalloidin for 60 min at room temperature. A mounting medium with DAPI (ab104139, Abcam) was used to visualize the nucleus.

Immunostaining was also conducted on cryosections that were prepared with the following procedure. Fresh tissues were fixed in 4% PFA for 12–24 hr at 4°C, dehydrated in 20% sucrose (in 1× PBS) for 2–3 hr followed by 30% sucrose (in 1× PBS) overnight at 4°C, embedded in OCT, and cut into cryosections at 8 µm thickness using a cryotome (CM1950, Leica Biosystems). Once sectioned, the samples were fixed on the slides with 4% PFA for 10–15 min at room temperature and washed three times with PBS. Subsequent steps were performed in the same way as paraffin sections were stained.

## Sperm count

Mice epididymal sperm were released into PBS through multiple incisions of the cauda followed by incubation for 30–60 min at 37°C under 5% $CO_2$. The living sperm was then diluted in PBS at 1:50 and transferred to a hemocytometer for counting. For all experiments, adult male mice 2–6 months of age were used.

## Protein extraction and western blot analysis

Testes from adult mice were cut into ~20 mg pieces and transferred into ~200 µl pre-cooled denaturing buffer from the Minute Total Protein Extraction Kit (SD-001, Invent Biotech, Plymouth, MN, USA) containing protease inhibitors (CO-RO, Roche, Indianapolis, IN, USA) and phosphatase inhibitors (PHOSS-RO, Roche). Protein extraction was performed with the Minute Total Protein Extraction Kit following the manufacturer's instructions. To load equal amounts of protein, the protein concentration in the supernatant was measured using the Pierce BCA Protein Assay Kit (23225, Thermo Fisher Scientific, Waltham, MA, USA). After diluting with loading buffer and boiling for 10 min at 95°C, an aliquot of 20 mg protein lysate per sample was resolved by sodium dodecyl sulfate-polyacrylamide gel electrophoresis, transferred to polyvinylidene fluoride membranes, blocked with noise-cancelling reagents (WBAVDCH01, Millipore, Burlington, MA, USA), and immunoblotted with the appropriate primary antibodies. The membranes were incubated with HRP-conjugated secondary antibodies followed by immunodetection using a chemiluminescence imaging system (5200, Tanon Technology, Shanghai, China).

## Surface chromosome spreading

Chromosome spread analysis was conducted with the drying-down technique as previously described (*Peters et al., 1997*). Briefly, the spermatocytes from hypotonicity-treated testicular tubules were

suspended in 0.1 M sucrose, spread on glass slides, and immersed in a PFA solution containing 0.15% Triton X-100. Subsequent immunolabeling of spermatocyte nuclei using antibodies against SYCP1 and SYCP3 was performed according to the protocol used for immunofluorescent staining of cryosections.

## Isolation of spermatogenic cells and FACS

Spermatogenic cells were isolated from the testes of 8-week-old male mice and subsequently subjected to flow cytometry and FACS for DNA ploidy analysis. In brief, testes collected from 8-week-old mice were decapsulated and washed in pre-chilled PBS three times. After carefully cutting, pieces of the seminiferous tubules were transferred into PBS containing 120 U/ml collagenase I (17018029, Gibco, Carlsbad, CA, USA) for a 5 min incubation at 32°C. To obtain cell suspensions, the tubular pieces were incubated in 0.25% trypsin (25200056, Gibco) containing 1 mg/ml DNase I (18047019, Thermo Fisher Scientific) at 32°C for 8 min with gentle pipetting. Cold DMEM containing 5% fetal bovine serum (10091155, Gibco) was added to stop the digestion. The spermatogenic cells were collected from the cell suspension after filtration through PBS-saturated 70 μm cell filters followed by centrifugation at 4°C (500×$g$ for 5 min). After removal of the supernatant, the cells were resuspended in 1 ml DMEM containing 2 μl Hoechst 33342 (62249, Thermo Fisher Scientific), 2 μl Zombie Aqua dye (423101, BioLegend, San Diego, CA, USA), and 5 μl DNase I and subjected to flow cytometry. For FACS, the cell suspension was further rotated at a speed of 10 rpm/min for 20 min at 32°C and centrifuged for 5 min at 4°C. Finally, the cells prepared for sorting were resuspended and cell populations were collected based on their fluorescent label with Hoechst 33342 staining using FACS.

## Electron microscopy

For TEM, testes collected from adult mice were cut into 1–2 mm$^3$ pieces and fixed in 2.5% glutaraldehyde in 0.1 M cacodylate buffer (pH 7.4) at 4°C overnight. After three washes in 0.1 M cacodylate buffer, the samples were incubated in 1% osmium tetroxide at room temperature for 1 hr. The samples were then washed with ultrafiltered water and stained in 2% uranyl acetate for 30 min. Subsequent dehydration was done through consecutive incubation in graded ethanol series (50%, 70%, 90%, and absolute ethanol) and an acetone bath. After dehydration, samples were sequentially incubated in Embed-812 resin: acetone mixtures at 1:1 and 3:1 for 2 hr each and in 100% Embed-812 resin for 3–4 hr, all at room temperature. The samples were then orientated and embedded with fresh resin at 37°C. Tissue blocks were cut on an ultramicrotome (UC7, Leica Biosystems) to yield 80 nm ultrathin sections, and these were subsequently contrasted with uranyl acetate and lead citrate. The sections were then observed and imaged using a TEM operating at 120 kV (JEM-1400, JEOL, Pleasanton, CA, USA).

## Mouse seminiferous tubule squashes

Seminiferous tubule squash slides were prepared as described previously (*Wellard et al., 2018*) with minor modifications. Specimens of testes were removed from adult mice and washed with PBS. The testicular tunica albuginea was torn, and the seminiferous tubules were released, collected, and fixed with 2 ml fixing solution for 5 min at room temperature in a 35 mm Petri dish. After washing twice with PBS, the seminiferous tubules were cut into small pieces (about 10–20 mm in length) that were subsequently transferred to the prepared glass slides containing 100 μl fixing/lysis solution. Coverslips were then applied to cover the glass slides and were compressed tightly for 20 s. After squashing, the glass slides were immediately frozen in liquid N$_2$ for 15 s and stored at –80°C. To perform immunolabeling on the squash slides, the coverslips were first removed after rewarming. The slides were immersed in PBS and washed three times. The subsequent steps were similar to the procedure for immunofluorescent staining of cryosections.

## Imaging

Immunolabeled slides, including paraffin sections, cryosections, squash slides, and chromosome spreads, were imaged by confocal microscopy (Dragonfly Spinning Disc confocal microscope driven by Fusion Software, Andor Technology, Belfast, UK). Projection images were then processed and analyzed using Bitplane Imaris (version 9.7) software. The histological and TUNEL-stained samples were imaged using an epifluorescence microscope (BX52, Olympus) equipped with a digital camera (DP80, Olympus) and processed using cellSens Standard (Olympus) software packages.

## Statistical analysis

Statistical analyses were conducted with SPSS software (version 22.0, IBM Corporation, Armonk, NY, USA). All data are presented as the mean ± SEM, as indicated in the figure legends. The statistical significance of the difference between the mean values for the various genotypes was determined by Welch's t-test with a paired two-tailed distribution. The data were considered significant based on p-value less than 0.05.

All data generated or analyzed during this study are included in the manuscript and the Source data files as: *Figure 1—source data 1*, *Figure 2—source data 1*, *Figure 3—source data 1*, *Figure 4—source data 1*, *Figure 5—source data 1*, *Figure 6—source data 1*, *Figure 7—source data 1*, *Figure 1—figure supplement 1—source data 1*, *Figure 1—figure supplement 2—source data 1*, *Figure 2—figure supplement 1—source data 1*, *Figure 2—figure supplement 2—source data 1*, *Figure 5—figure supplement 1—source data 1*, *Figure 7—figure supplement 1—source data 1*.

## Acknowledgements

We thank our colleagues at the Center for Reproductive Medicine, Shandong University, for their technical support. Ssh2-C-HA mice were supplied by GTP Center, CEMCS, CAS, which was supported by Shanghai Municipal Commission for Science and Technology Grants (19411951800). This work was supported by the National Key R&D Program of China (2022YFC2702600 to HL and 2021YFC2700500 to JM); and the Chinese Academy of Medical Sciences (2020RU001 to Z-JC); and the Academic Promotion Programme of Shandong First Medical University (2019U001 to Z-JC); and the National Natural Science Foundation of China (82071699 and 81771538 to HL); and the Major Innovation Projects in Shandong Province (2021ZDSYS16 to HL); and the Science Foundation for Distinguished Young Scholars of Shandong (ZR2021JQ27 to HL); and the Taishan Scholars Program for Young Experts of Shandong Province (tsqn202103192 to HL); and the Basic Science Center Program of NSFC (31988101 to Z-JC); and the Shandong Provincial Key Research and Development Program (2020ZLYS02 to Z-JC). The funders had no role in study design, data collection, and interpretation, or the decision to submit the work for publication.

## Additional information

### Funding

| Funder | Grant reference number | Author |
|---|---|---|
| National Key Research and Development Program of China | 2021YFC2700500 | Jinlong Ma |
| National Key Research and Development Program of China | 2022YFC2702600 | Hongbin Liu |
| Chinese Academy of Medical Sciences | 2020RU001 | Zi-Jiang Chen |
| Shandong First Medical University | Academic Promotion Programme | Zi-Jiang Chen |
| Major Scientific and Technological Innovation Project of Shandong Province | 2021ZDSYS16 | Hongbin Liu |
| Natural Science Fund for Distinguished Young Scholars of Shandong Province | ZR2021JQ27 | Hongbin Liu |
| Taishan Scholar Project of Shandong Province | Program for Young Experts | Hongbin Liu |

| Funder | Grant reference number | Author |
|--------|------------------------|--------|
| National Natural Science Foundation of China | Basic Science Center Program | Zi-Jiang Chen |
| Key Technology Research and Development Program of Shandong | 2020ZLYS02 | Zi-Jiang Chen |
| National Natural Science Foundation of China | 82071699 | Hongbin Liu |
| National Natural Science Foundation of China | 81771538 | Hongbin Liu |
| Shandong First Medical University | 2019U001 | Zi-Jiang Chen |
| Taishan Scholar Project of Shandong Province | tsqn202103192 | Hongbin Liu |
| National Natural Science Foundation of China | 31988101 | Zi-Jiang Chen |

The funders had no role in study design, data collection and interpretation, or the decision to submit the work for publication.

### Author contributions

Ke Xu, Data curation, Formal analysis, Investigation, Visualization, Methodology, Writing - original draft, Writing – review and editing; Xianwei Su, Data curation, Formal analysis, Validation, Investigation, Methodology, Writing – review and editing; Kailun Fang, Resources, Software, Methodology; Yue Lv, Formal analysis, Validation; Tao Huang, Shangming Liu, Xiangfeng Chen, Formal analysis; Mengjing Li, Yingying Yin, Methodology; Ziqi Wang, Investigation; Tahir Muhammad, Writing – review and editing; Jing Jiang, Jinsong Li, Resources; Wai-Yee Chan, Supervision; Jinlong Ma, Supervision, Funding acquisition; Gang Lu, Conceptualization, Supervision; Zi-Jiang Chen, Conceptualization, Supervision, Funding acquisition, Project administration; Hongbin Liu, Conceptualization, Data curation, Supervision, Funding acquisition, Project administration, Writing – review and editing

### Author ORCIDs

Kailun Fang ⓘ http://orcid.org/0000-0003-2456-6423
Tao Huang ⓘ http://orcid.org/0000-0002-7086-570X
Jinsong Li ⓘ http://orcid.org/0000-0003-3456-662X
Zi-Jiang Chen ⓘ http://orcid.org/0000-0001-6637-6631
Hongbin Liu ⓘ http://orcid.org/0000-0003-2550-7492

### Ethics

All procedures were in accordance with the ethical standards approved by the Animal Use Committee of the School of Medicine, Shandong University, for the care and use of laboratory animals. The research was approved by the Institutional Review Board of Shandong University. (Reference Number: 2019#057).

### Decision letter and Author response

Decision letter https://doi.org/10.7554/eLife.83129.sa1
Author response https://doi.org/10.7554/eLife.83129.sa2

## Additional files

### Supplementary files
• MDAR checklist

### Data availability

All data generated or analysed during this study are included in the manuscript and supporting file; Source Data files have been provided for Figures 1-7 and Figure supplement.

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
