## [Editor Report]

This important study reports convincing data supporting the essential role of SSH2 in the regulation of acrosome biogenesis during spermiogenesis. The conclusion is well supported by the experiments performed both in vivo and in vitro. This work will help understand the molecular process of sperm assembly and also identify potential genetic mutations responsible for acrosomal defects in male infertility patients.

---

## [Decision Letter]

**Decision letter after peer review:**

Thank you for submitting your article "The Slingshot phosphatase 2 is required for acrosome biogenesis during spermatogenesis in mice" for consideration by *eLife*. Your article has been reviewed by 2 peer reviewers, and the evaluation has been overseen by a Reviewing Editor and Ricardo Azziz as the Senior Editor. The following individual involved in the review of your submission has agreed to reveal their identity: Sue Hammoud (Reviewer #1).

This work revealed the physiological role of SSH2 in the regulation of spermatogenesis. Both phenotypic and molecular characterization was well conducted, and the data are generally convincing. Only some minor concerns were identified and need to be addressed in a revised version.

Essential revisions:

1) Provide Phalloidin staining on tubule squashes to examine actin cytoskeleton.

2) Provide EM analyses of the round spermatids to detect the cytoplasmic distribution of actin filaments in the wild type and disrupted actin filament remodeling in KO mice, if possible.

*Reviewer #2 (Recommendations for the authors):*

Nice work and it could enhance the impact of this work by examining carefully the defect of actin cytoskeleton remodeling, proacrosomal vesicle trafficking, or any other subcellular organelles in round spermatids with EM. It is important to distinguish the general effect of the actin cytoskeleton and specific effects on acrosomal biogenesis such as defects in Golgi and proacrosomal vesicle fusion.

---

## [Author Response]

Essential revisions:1) Provide Phalloidin staining on tubule squashes to examine actin cytoskeleton.

We appreciate the insightful comment by the editor and apologize for the lack of flourescent stain marks. We indicated the actin cytoskeleton which is the rhodamine phalloidin in the panels of Figure 5B in the originally submitted manuscript, namely “the representative images of Phalloidin staining performed on tubule squashes”. To clarify, we have now highlighted the mark of Phalloidin and added some indicators regarding the structures of disorganized F-actin in the images of a new Figure 5B in the revised manuscript. Furthermore, we have replaced the images of Phalloidin staining on tubule squashes with more representative ones in Figure 5B in the revised manuscript which demonstrated clearly that actin filament remodeling is disrupted in the round spermatids of *Ssh2* KO mice and solidified the conclusion that the actin remodeling was disordered in *Ssh2* KO spermatids.

2) Provide EM analyses of the round spermatids to detect the cytoplasmic distribution of actin filaments in the wild type and disrupted actin filament remodeling in KO mice, if possible.

Thank you for your comment. Previously we already have conducted a transmission electron microscopy (TEM) analysis on the testes samples to discover the distribution and ultrastructural organization of F-actin in WT and *Ssh2* KO round spermatids. Unfortunately, even at high magnification (30,000x, right panel of Author response image 1) by TEM of the testicular section no diminutive actin filament was observed in the cytoplasm of round spermatids except for the acroplaxome-an actin-rich specialized structure anchors the acrosome-in WT spermatids as well as some thick bundle-like structures located at the acrosomal region of *Ssh2* KO spermatids (Author response image 1). According to their unique characteristic of appearance, we interpreted these electron-dense bundles as the aberrantly aggregated actin filaments whose lengths are in accordance with the lengths of COFILIN-saturated F-actin fragments (Bamburg, Minamide, Wiggan, Tahtamouni, and Kuhn, 2021), suggesting the disrupted actin filament remodeling during acrosome biogenesis resulted from *Ssh2* KO. However, due to the technological limitations of TEM and the complexity of the intracellular environment of round spermatids, we only recognized a few aggregated actin bundles with the loss of filamentous appearance in *Ssh2* KO spermatids and no typical diminutive actin filament was detected which had been imaged under high-resolution cryo-TEM (Haviv, Gov, Ideses, and Bernheim-Groswasser, 2008) or live-cell total internal reflection fluorescence microscopy (Johnson et al., 2015) on the purified actin bundles and cultured cells. Given the lack of effective approaches to culture murine round spermatids in vitro, confocal microscopy of flourescence-labelled F-actin (*e.g.*, IF staining by FITC-phalloidin) is a more accessible method for visualizing the disruption of actin remodeling than EM in murine spermatids as the actin-related findings that several other studies demonstrated (Djuzenova et al., 2015).

**Author response image 1. sa2fig1:** Representative images of ultrastructural analysis of actin organization in WT and *Ssh2* KO spermatids. No intact F-actin structure was observed in WT round spermatids with few aggregated actin bundles (as red arrowheads indicated) detected in *Ssh2* KO spermatids under TEM. The regions outlined by white boxes are shown at higher magnification to the right. Nu, nucleus; Ac, acrosome; Golgi, Golgi apparatus; Apx, acroplaxome; Pav, proacrosomal vesicle. Scale bars: left panel, 1 µm; right panel, 0.5 µm.

Reviewer #2 (Recommendations for the authors):Nice work and it could enhance the impact of this work by examining carefully the defect of actin cytoskeleton remodeling, proacrosomal vesicle trafficking, or any other subcellular organelles in round spermatids with EM. It is important to distinguish the general effect of the actin cytoskeleton and specific effects on acrosomal biogenesis such as defects in Golgi and proacrosomal vesicle fusion.

We appreciate and thank the referee for the positive feedback which helps us to keep these suggestions in our mind to perform the experiments in our future projects.

References:

Bamburg, J. R., Minamide, L. S., Wiggan, O., Tahtamouni, L. H., and Kuhn, T. B. (2021). Cofilin and Actin Dynamics: Multiple Modes of Regulation and Their Impacts in Neuronal Development and Degeneration. *Cells, 10*(10). doi:10.3390/cells10102726

Clermont, Y., and Leblond, C. P. (1955). Spermiogenesis of man, monkey, ram and other mammals as shown by the periodic acid-Schiff technique. *Am J Anat, 96*(2), 229-253. doi:10.1002/aja.1000960203

Djuzenova, C. S., Fiedler, V., Memmel, S., Katzer, A., Hartmann, S., Krohne, G.,... Sukhorukov, V. L. (2015). Actin cytoskeleton organization, cell surface modification and invasion rate of 5 glioblastoma cell lines differing in PTEN and p53 status. *Exp Cell Res, 330*(2), 346-357. doi:10.1016/j.yexcr.2014.08.013

Endo, M., Ohashi, K., and Mizuno, K. (2007). LIM kinase and slingshot are critical for neurite extension. *J Biol Chem, 282*(18), 13692-13702. doi:10.1074/jbc.M610873200

Haviv, L., Gov, N., Ideses, Y., and Bernheim-Groswasser, A. (2008). Thickness distribution of actin bundles in vitro. *Eur Biophys J, 37*(4), 447-454. doi:10.1007/s00249-007-0236-1

Huang, Q., Liu, Y., Zhang, S., Yap, Y. T., Li, W., Zhang, D.,... Zhang, Z. (2021). Autophagy core protein ATG5 is required for elongating spermatid development, sperm individualization and normal fertility in male mice. *Autophagy, 17*(7), 1753-1767. doi:10.1080/15548627.2020.1783822

Johnson, H. E., King, S. J., Asokan, S. B., Rotty, J. D., Bear, J. E., and Haugh, J. M. (2015). F-actin bundles direct the initiation and orientation of lamellipodia through adhesion-based signaling. *J Cell Biol, 208*(4), 443-455. doi:10.1083/jcb.201406102

Khawar, M. B., Gao, H., and Li, W. (2019). Mechanism of Acrosome Biogenesis in Mammals. *Front Cell Dev Biol, 7*, 195. doi:10.3389/fcell.2019.00195

Meenderink, L. M., Gaeta, I. M., Postema, M. M., Cencer, C. S., Chinowsky, C. R., Krystofiak, E. S.,... Tyska, M. J. (2019). Actin Dynamics Drive Microvillar Motility and Clustering during Brush Border Assembly. *Dev Cell, 50*(5), 545-556 e544. doi:10.1016/j.devcel.2019.07.008

Meistrich, M. L., and Hess, R. A. (2013). Assessment of spermatogenesis through staging of seminiferous tubules. *Methods Mol Biol, 927*, 299-307. doi:10.1007/978-1-62703-038-0_27

Nakata, H., Wakayama, T., Takai, Y., and Iseki, S. (2015). Quantitative analysis of the cellular composition in seminiferous tubules in normal and genetically modified infertile mice. *J Histochem Cytochem, 63*(2), 99-113. doi:10.1369/0022155414562045

Teves, M. E., Roldan, E. R. S., Krapf, D., Strauss, J. F., III, Bhagat, V., and Sapao, P. (2020). Sperm Differentiation: The Role of Trafficking of Proteins. *Int J Mol Sci, 21*(10). doi:10.3390/ijms21103702

Wang, H., Wan, H., Li, X., Liu, W., Chen, Q., Wang, Y.,... Li, W. (2014). Atg7 is required for acrosome biogenesis during spermatogenesis in mice. *Cell Res, 24*(7), 852-869. doi:10.1038/cr.2014.70

Xiao, N., Kam, C., Shen, C., Jin, W., Wang, J., Lee, K. M.,... Xia, J. (2009). PICK1 deficiency causes male infertility in mice by disrupting acrosome formation. *J Clin Invest, 119*(4), 802-812. doi:10.1172/JCI36230